# Rebels with a cause? How norm violations shape dominance, prestige, and influence granting

Gerben A. van Kleef\*, Florian Wanders¤a, Annelies E. M. van Vianen, Rohan L. Dunham, Xinkai Du¤b, Astrid C. Homan

Department of Psychology, University of Amsterdam, Amsterdam, The Netherlands

¤a Current address: Booking.com, Amsterdam, The Netherlands
¤b Current address: University of Oslo, Oslo, Norway
\* g.a.vankleef@uva.nl

**Data Availability Statement:** All data have been uploaded onto a publicly accessible repository. The data can be accessed here: https://osf.io/f42wq/.

## Abstract

Norms play an important role in upholding orderly and well-functioning societies. Indeed, violations of norms can undermine social coordination and stability. Much is known about the antecedents of norm violations, but their social consequences are poorly understood. In particular, it remains unclear when and how norm violators gain or lose influence in groups. Some studies found that norm violators elicit negative responses that curtail their influence in groups, whereas other studies documented positive consequences that enhance violators' influence. We propose that the complex relationship between norm violation and influence can be understood by considering that norm violations differentially shape perceptions of dominance and prestige, which tend to have opposite effects on voluntary influence granting, depending on the type of norm that is violated. We first provide correlational (Study 1) and causal (Study 2) evidence that norm violations are associated with dominance, and norm abidance with prestige. We then examine how dominance, prestige, and resultant influence granting are shaped by whether local group norms and/or global community norms are violated. In Study 3, protagonists who violated global (university) norms but followed local (sorority/fraternity) norms were more strongly endorsed as leaders than protagonists who followed global norms but violated local norms, because the former were perceived not only as high on dominance but also on prestige. In Study 4, popular high-school students were remembered as violating global (school) norms while abiding by local (peer) norms. In Study 5, individuals who violated global (organizational) norms while abiding by local (team) norms were assigned more leadership tasks when global and local norms conflicted (making violators "rebels with a cause") than when norms did not conflict, because the former situation inspired greater prestige. We discuss implications for the social dynamics of norms, hierarchy development, and leader emergence.

**Funding:** The author(s) received no specific funding for this work.

## Introduction

Human societies are held together by social norms, implicit or explicit rules or principles that are understood by members of a group and that guide and/or constrain behavior [1]. By creating a shared understanding of what is acceptable within a particular context and what is not, social norms enable coordinated action in human collectives across society, from school classes to organizational departments to international politics [2, 3]. Because norms are critical to the functioning of groups and communities, much research in recent decades has focused on the influence of norms on human behavior [4–6]. Whereas these efforts have yielded great insight in the mechanics of social norms, comparatively little is known about the social dynamics that ensue when norms are violated. In particular, the consequences of norm violations for norm violators' positions in social collectives remain poorly understood. Anecdotally, news reports about the career trajectories of celebrities, CEOs, and politicians suggest that some fall from grace after a transgression, whereas others gain popularity and influence. This raises the question of how norm violations shape transgressors' social standing, which we set out to address in the current research.

Previous research that speaks to this question has yielded inconclusive results. On the one hand, several studies found that norm violators elicited unfavorable responses from others, such as negative emotions [7–10], gossip [11, 12], social exclusion [13, 14], and various forms of social or monetary punishment [15–18]. Such responses discourage future transgressions [13, 19, 20] and undermine norm violators' standing and influence in groups [10, 21]. On the other hand, research indicates that norm violators may enjoy advantageous social consequences, such as enhanced popularity [22, 23], perceived power [10, 24, 25], status [26], and artistic impact [27]. Thus, norm violators may gain standing and influence in groups.

These inconsistent findings reflect that the social consequences of norm violations are complex, multi-faceted, and subject to moderating influences. Indeed, previous research has shown that responses to norm violations are moderated by characteristics of the actor, the observer, and the cultural context [28]. Research examining actor characteristics, for instance, has found that norm violators are less likely to be met with punishment to the degree that they have greater power [29]. Conversely, studies on observer characteristics have found that negative responses to norm violations become more likely to the degree that observers feel more powerful or have greater social-economic status [30]. Finally, cross-cultural investigations revealed that negative responses to norm violations are more prevalent in comparatively collectivistic and tight cultures than in individualistic and loose cultures [10]. In the current research we seek to gain further insight in the social dynamics of counternormative behavior by considering the nature of the norms involved.

Below, we begin by making the basic argument that social norms are heavily contextualized and vary across social collectives [5, 31]. Given that people are a part of numerous social collectives, they may be confronted with conflicting normative expectations. How people navigate such normative tensions, we propose, has implications for their social-hierarchical standing. To illuminate these implications, we draw on the dominance/prestige framework of social rank [32–34], which we use to advance the novel hypothesis that norm violations generally increase actors' perceived dominance, whereas they decrease their prestige. We then develop this idea further by arguing that norm violators' prestige shifts depending on whether global community norms or local group norms are violated and whether community norms and group norms do or do not conflict with each other. From this model we derive a series of specific predictions, which we test in five studies using a multi-method approach. We conclude by considering implications of our findings, discussing strengths and limitations of the current approach, and suggesting avenues for future research.

## The contextualized nature of social norms

Although norms universally support the functioning of social collectives [2, 3], the nature of these norms varies across groups and cultures as a function of situational demands and locally endorsed values [5, 31, 35]. For instance, different extended families may have different norms concerning the frequency with which they meet, different organizations may have different norms regarding appropriate work attire, different universities may have different norms about publishing articles versus books, and different groups of friends may have different norms pertaining to alcohol consumption. It follows that in everyday social situations people may be confronted with multiple, potentially conflicting, norms, such that abidance by a norm that is endorsed in one group entails violation of a norm that is endorsed in another group [36]. Such conflicts between different normative systems may not be all that salient when groups have no physical proximity or overlap in their members, but they become apparent when groups intersect, such as when one (smaller) group is part of another (larger) group.

Much of human life is organized in nested collectives, with individuals belonging to groups that are in turn embedded in larger social structures [37]–consider a group of students nested in a school, a lab group embedded in a university department, a political faction in parliament, or a project team working within a larger company. Given that norms can vary widely between different social collectives [31], the norms that are endorsed at the local group level may or may not be compatible with the norms that are espoused at the global community level. For instance, in a consultancy company, norms regarding acceptable work attire are compatible between levels if the IT department (local group level) and the company at large (global community level) endorse the same formal dress code, but incompatible if the IT department embraces a more leisurely dress code than the company at large prescribes. Similarly, in a political context, norms pertaining to language use and interpersonal comportment may be aligned or misaligned between political factions and parliament as a whole.

When faced with conflicting normative expectations, people are forced to prioritize one set of norms over another. For instance, an employee of the IT department of the consulting firm must decide whether to abide by her local group's casual dress norms and violate the firm's formal dress code, or vice versa. This decision may have notable consequences for how the employee is regarded by others, because social norms have great symbolic meaning for groups [38, 39]. Besides facilitating coordination, social norms capture distinctive properties of groups that may be part of the group's identity [40]. Thus, abidance by group norms not only facilitates attainment of group goals, it also signals loyalty and commitment to the group. Critically, however, in situations in which local group norms conflict with global community norms, it is impossible for group members to abide by both sets of norms at the same time. This begs the question of how decisions to abide by or violate one or the other set of norms shape people's standing and influence in social collectives. Below we draw on the dominance/prestige framework of social rank to shed light on this question.

## Dominance and prestige as distinct routes to rank and influence

The dominance/prestige framework of social rank posits that social hierarchies are shaped by two overarching systems of rank allocation: dominance and prestige [33]. Dominance and prestige are commonly conceptualized as two different "strategies" for attaining social rank [32], which play distinct roles in the dynamic regulation of status and influence in social hierarchies [32, 34, 41–43].

Dominance refers to behaviors that bolster one's position in a social hierarchy through force or threat of force [32, 33]. A typical dominance strategy, as seen in humans as well as numerous non-human animals, involves assertiveness, aggression, coercion, intimidation, or

the use of punishment and reward to gain or maintain influence. A typical response to such behaviors is passive submissiveness, fueled by fear. Although coming across as dominant helps individuals to forcefully claim influence through intimidation, it is less often effective when it comes to eliciting voluntary granting of leadership and influence among humans [41, 42], because dominant individuals tend to be disliked by and evoke negative reactions in interaction partners [44, 45]. Preferences for dominant individuals as leaders or allies do increase under particular circumstances, such as economic uncertainty [43] and intergroup competition [46–49]. Generally speaking, however, people appear to be reluctant to voluntarily grant influence to individuals who display dominance [50, 51].

Such voluntary granting of influence is more commonly observed in connection with prestige, which is thought to have evolved as a psychological adaptation to highlight individuals who possess attributes and dispositions that generate benefits for the group [33]. A typical prestige strategy involves the display of locally valued skills, expertise, abilities, and knowledge to garner respect and, ultimately, freely conferred rank and influence [32, 34]. Research has shown that proximal prestige-related characteristics such as intelligence, competence, and expertise [44, 52–54] and more distal yet associated characteristics such as group commitment, ambassadorship, self-sacrifice, and prosociality [55–59] predict group members' perceived value to the group, the status they are accorded by fellow group members, and their actual influence on group decisions. Thus, people can attract endorsement as leaders and obtain influence in groups by amassing prestige.

In sum, there is evidence that dominance and prestige play distinct roles in the allocation of rank and the emergence and maintenance of social hierarchies [32–34, 42, 43, 49]. Individuals who display dominance may be allowed influence because they evoke intimidation and fear, but they tend not to receive voluntarily granted leadership. Individuals who display prestige, in contrast, earn freely granted leadership and influence through respect and admiration [50]. Thus, all else being equal, people are less likely to be voluntarily granted influence to the degree that they are perceived as dominant, whereas they are more likely to receive freely granted influence to the degree that they are perceived as prestigious.

## Norm violation versus abidance as cues of dominance versus prestige

Given that perceptions of dominance and prestige shape people's willingness to grant influence to others, taking these perceptions into account can enhance understanding of whether and how norm violators gain influence. Theoretical arguments and suggestive evidence provide a basis for hypotheses about effects of norm violations on both dominance and prestige.

How norm violations shape perceptions of dominance. Violating norms can elicit negative responses from bystanders. To the degree that bystanders value prevailing norms, norm violators may face pushback from others when they undermine those norms [19]. Specifically, norm violators risk being sanctioned [25], for instance via reprimands [60], social exclusion [14], or monetary fines [18]. People who appear to knowingly and intentionally violate norms signal that they are willing and able to withstand such backlash. The capacity to withstand backlash and prevail in confrontations is a characteristic of dominant individuals [61], who are prepared to use (threat of) force when challenged [32, 62]. Thus, by virtue of the fact that norm violations entail risk of negative repercussions that dominant individuals are more willing and able to bear, norm violations may fuel perceptions of dominance. Suggestive evidence for this possibility comes from research indicating that individuals who violated rules of bookkeeping, stole coffee from service personnel, or dropped their cigarette ashes on the floor despite the presence of an ashtray were perceived as occupying more powerful positions than individuals who behaved normatively [24]. Clearly, a person's power position cannot be

equated with their degree of dominance [52], but this finding is compatible with our theoretical argument that norm violations signal a person's capacity to withstand potential backlash. On a more general level, this reasoning is also in keeping with costly signaling theory, which posits that behaviors that are more costly to perform constitute more reliable signals of underlying qualities [63, 64]–in this case the capacity to weather possible sanctions, and hence dominance. We therefore hypothesized that norm violators are perceived as more dominant than norm abiders (*Hypothesis 1*).

**How norm violations shape perceptions of prestige.** A common explanation for the emergence of norms in social collectives is that they enable coordinated action [2, 3, 10]. To the degree that norms are thus deemed beneficial for the functioning of collectives, norm violators are likely to be perceived as a hindrance [19]. Indeed, common responses to norm violations, such as negative emotions [7, 10], gossip [11, 12], derogation [17], and social exclusion [14], suggest that norm violators are perceived as disruptive. Related research on opinion deviates has found that group members who espouse different attitudes or judgments than the majority of the group does are perceived by the majority as less valuable to the group [57] and often incur pushback from fellow group members who strive to protect or restore the group's positive identity, distinctiveness, cohesion, or effectiveness [65]. Such perceptions of deviants as offering little value to the group represent the opposite of prestige, which tracks the perceived value or benefits a person offers to the collective [32, 33]. We therefore predicted that, generally, norm violators are perceived as less prestigious than norm abiders (*Hypothesis 2*).

## When local group norms and global community norms conflict

When group and community norms are compatible, individuals can simultaneously abide by (or violate) norms at both levels. However, when norms at one level conflict with those at the other level, individuals cannot follow both norms at the same time. We propose that norm violations that ensue in such situations have different implications for actors' social-hierarchical positions in groups depending on whether they violate local group norms or global community norms, because the decision to prioritize local group norms versus global community norms affects a person's prestige in the eyes of fellow group members.

As noted above, prestige is thought to have evolved as a mechanism for highlighting individuals who possess attributes and dispositions that generate benefits for the group [33]. To the degree that abiding by norms that are valued by the group is perceived as beneficial to the group [5], individuals can be expected to accrue comparatively more prestige in the eyes of fellow group members by adhering to local group norms than by adhering to global community norms that may have less immediate implications for the group's functioning. This argument is compatible with the notion of the social identity perspective that group norms can become an important part of groups' identities [40]. Furthermore, indirect support for the argument comes from research in the realm of the subjective group dynamics model, which indicates that people respond more strongly to the (counter)normative behavior of ingroup rather than outgroup members [17, 66]. In light of these considerations, we hypothesized that when faced with a conflict between local group norms and global community norms, individuals gain more prestige among fellow group members when they choose to follow local group norms and violate global community norms than vice versa (*Hypothesis 3*).

## When norm violators are granted influence

The hypothesized effects of norm violations on perceived dominance and prestige, if supported, would shed fresh light on inconsistent past results concerning observers' tendencies to afford influence to (or withhold influence from) norm violators. The foregoing argument

suggests that norm violations generally undermine voluntary influence granting because they convey high dominance and low prestige. We have also argued, however, that perceptions of prestige may shift depending on whether a person violates local group norms or global community norms when the two are misaligned. If actors who abide by group norms while violating conflicting community norms indeed amass more prestige within their group than actors who exhibit the opposite pattern (as per Hypothesis 3), and assuming that perceptions of dominance are similar in both cases, we would expect fellow group members to be more willing to voluntarily grant influence to individuals who violate community norms while abiding by (conflicting) group norms than to individuals who violate group norms while abiding by (conflicting) community norms (*Hypothesis 4*). Furthermore, we hypothesized that this tendency to preferentially grant influence to individuals who violate community norms rather than group norms is mediated by prestige (*Hypothesis 5*).

Finally, our theoretical argument implies that group members' willingness to grant influence to individuals who violate community norms while abiding by group norms depends on whether group and community norms do or do not conflict with each other. This is because the hypothesized boost in prestige of individuals who violate community norms while abiding by group norms stems from the fact that these individuals apparently prioritized local group norms over global community norms. Given that prestige is conferred to individuals who benefit the group [33], they would likely be excused for violating community norms if doing so was necessary to uphold the norms of the group. In fact, violating community norms for the sake of respecting group norms can be seen as a way of demonstrating one's allegiance to the group [38, 40], which would likely translate in enhanced prestige. However, it follows from the general negative relationship between norm violation and prestige hypothesized above that, when group and community norms can both be respected at the same time and violating community norms is therefore not necessary to abide by group norms, individuals who nonetheless violate community norms would be perceived as less prestigious and would consequently be less likely to be granted influence. Thus, we predicted that individuals who violate community norms while abiding by group norms are perceived as more prestigious by fellow group members when local group norms and global community norms conflict–that is, when violators rebel with a cause–than when they do not conflict (*Hypothesis 6*). Accordingly, we hypothesized that individuals who violate community norms while abiding by group norms are more likely to be granted influence when local group norms and global community norms conflict than when they do not conflict (*Hypothesis 7*). Finally, we hypothesized that this greater willingness to grant influence to individuals who violate community norms while abiding by group norms when the two conflict is mediated by prestige (*Hypothesis 8*).

## Overview of the present studies

We tested our hypotheses in five studies, using a variety of samples, settings, and methods. In Study 1, we employed an implicit association test (IAT) to examine whether people generally associate norm violations with dominance and norm abidance with prestige. In Study 2, we manipulated an actor's (counter)normative behavior in a car parking context to examine causal effects of norm violation versus abidance on perceived dominance and prestige. In Study 3, we used a university graduation ceremony context to examine perceived dominance and prestige and concomitant influence granting as a function of an actor's violation of or abidance by a local (fraternity/sorority) and/or a global (university) dress norm. In this study, we operationalized influence granting by means of (intersubjective) leadership support tendencies. In Study 4, we employed a critical-incident recall paradigm to examine whether the popularity of high-school students (as a proxy of their influence) is associated with their

recalled violation of or abidance by global (school) norms versus local (peer) norms. Finally, in Study 5, we examined whether people are more willing to grant influence to organization members who prioritize local group norms over global community norms when the two norms conflict than when they do not conflict, because the former situation would result in greater prestige for the violator than the latter. In this study, we operationalized influence granting through a behavioral measure involving the assignment of leadership tasks to the norm violator.

For all studies, sample sizes and exclusion criteria were determined a priori, and data exclusions (if any) are explicitly indicated and justified. The minimum sample size of each study was determined based on *a priori* power analysis, as detailed in each individual study. In all studies, we oversampled to account for possible participant dropout (e.g., due to technical malfunction, non-compliance, duplicate responses, or lack of understanding). We also provide sensitivity power analyses for each study. All study protocols were approved by the local ethics review board, as detailed in the individual studies. All measures and manipulations are reported. Data files and code of all studies are available here: https://osf.io/f42wq/.

## Study 1

The goal of Study 1 was to establish whether people generally associate norm violations with dominance and norm abidance with prestige, as suggested by Hypotheses 1 and 2, respectively. An established procedure for uncovering such associations is the implicit association test (IAT), which assesses the strength of associations between different concepts or categories [67]. In an IAT, category labels are displayed on either side of the computer screen (see Fig 1). Participants are asked to categorize target words appearing in the middle of the screen as belonging either to the categories on the left side or to the categories on the right side as quickly and accurately as possible. A strong association between two categories or concepts (e.g., norm violation and dominance) results in reduced reaction times and fewer errors when they are presented on the same side of the screen (as in the left panel of Fig 1) rather than on different sides (as in the right panel of Fig 1). Although the IAT is best known for its use in research on implicit prejudice and other valenced associations, it can be used to study any type of implicit association, including associations with concepts related to power and leadership [e.g., 49, 68]. Here we used the IAT to examine implicit associations between norm violation versus abidance and dominance versus prestige.

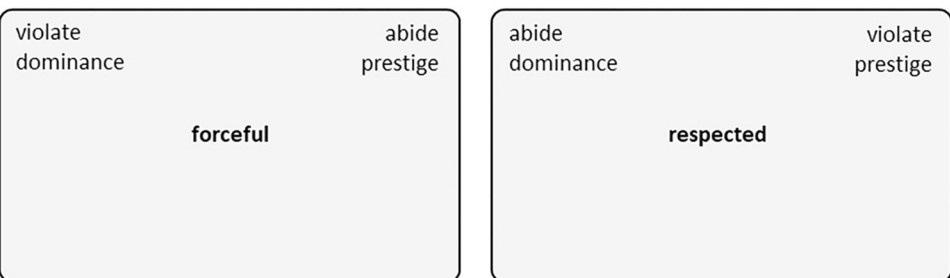

**Fig 1. Example screens from the implicit association test (IAT) used to assess associations between norm violation versus abidance and dominance versus prestige (Study 1).** Across multiple trials (see Table 1), participants' task was to categorize target words related to norm violation, norm abidance, dominance, and prestige that appeared in the center of the screen (e.g., forceful, respected) as belonging to the categories on the left side or the categories on the right side of the screen as quickly and accurately as possible.

## Methods

**Participants and design.**   We used G*Power [69] to determine the minimum sample size needed to detect a medium-sized effect ($d = 0.5$) in a paired-samples $t$ test (reflecting the within-participants nature of the IAT setup), with a significance level of $\alpha = .05$ and a power of .90. The analysis recommended a minimum sample size of 36 participants. We exceeded this number to accommodate possible loss of data due to computer malfunction or unreliable responses [70; see below] and to enhance statistical power. We recruited 135 undergraduate students from a large university in the Netherlands (104 women, 31 men; $M_{age} = 21.76$, range: 18–52). Sensitivity analysis indicated that, given a conventional significance level of $\alpha = .05$ (two-tailed) and a standard power criterion of 80%, this sample size allowed for detecting an effect size of $d = 0.24$ or greater (i.e., a small- to medium-sized effect).

All participants were presented with the same stimuli, which were varied in a (counterbalanced) block-wise fashion in a within-participants design, as detailed below. The study was conducted at the research laboratory of the psychology department of the University of Amsterdam, where participants completed the study on dedicated computers configured to register response latencies with the greatest possible accuracy. Participants were compensated with course credit or monetary reimbursement (€2.00, approx. $2.40). The study was approved by the Ethics Review Board of the Faculty of Social and Behavioral Sciences of the University of Amsterdam (case number: 2016-WOP-6822). Data were collected between 25 April 2016 and 31 May 2016. All participants provided written informed consent.

**Procedure and materials.**   Dutch stimulus words for the IAT were developed with the help of an online thesaurus and piloted in a separate sample by asking participants to assign each stimulus word into one of the four categories of norm abidance, norm violation, dominance, and prestige ($N = 26$; 19 women, 7 men; $M_{age} = 22.77$, range: 19–27). The words that were most consistently assigned to the correct category were selected for use in the actual study (the original Dutch words are reported in the Supporting Information file). This procedure ensured that the selected words were understood as intended. For *norm abidance*, we used obedient, follow, comply, accommodating, and well-behaved; for *norm violation*, we used unruly, violate, rebellious, transgress, and naughty; for *dominance*, we used domineering, bossy, forceful, dominance, and authoritarian; and for *prestige*, we used admired, status, prestigious, respected, and honorable.

The IAT task was set up following the procedure recommended by Greenwald and colleagues [70, 71]. The task consisted of seven blocks (see Table 1), in which participants were instructed to categorize stimulus words appearing in the middle of the screen (e.g., forceful, respected) as belonging to either the categories shown on the left side of the screen or to those

**Table 1. Overview of the seven blocks of the implicit association test (IAT) used to assess associations between norm violation versus abidance and dominance versus prestige (Study 1).**

| Block | No. of Trials | Function | Categories | Items assigned to left key response ("Q") | Items assigned to right key response ("P") |
|---|---|---|---|---|---|
| 1 | 20 | Practice | Dominance vs. prestige | Dominance | Prestige |
| 2 | 20 | Practice | Norm violation vs. abidance | Norm violation | Norm abidance |
| 3 | 20 | Test | Combined | Dominance + norm violation | Prestige + norm abidance |
| 4 | 40 | Test | Combined | Dominance + norm violation | Prestige + norm abidance |
| 5 | 40 | Practice | Dominance vs. prestige (reversed) | Prestige | Dominance |
| 6 | 20 | Test | Combined (reversed) | Prestige + norm violation | Dominance + norm abidance |
| 7 | 40 | Test | Combined (reversed) | Prestige + norm violation | Dominance + norm abidance |

Left (Q) and right (P) key assignment and the order of Blocks 3 and 4 and Blocks 6 and 7 were counterbalanced.

shown on the right side of the screen (see Fig 1) by pressing the Q or P key on the computer keyboard, respectively.

After two practice blocks with two categories (dominance and prestige in Block 1, norm violation and norm abidance in Block 2), participants completed two test blocks (Blocks 3 and 4) with four categories (two on each side of the screen). If people hold strong associations between two categories, assigning stimulus words to categories is easier (manifested in greater speed and accuracy) when categories are shown on the same side of the screen (congruent trials) compared to when they are shown on different sides of the screen (incongruent trials). For half of the participants, Blocks 3 and 4 were congruent trials, in which the categories dominance and norm violation were presented on one side of the screen and the categories prestige and norm abidance were presented on the other side; for the other half of the participants, Blocks 3 and 4 were incongruent trials, in which the categories dominance and norm abidance were presented on one side of the screen and the categories prestige and norm violation were presented on the other side.

To avoid confusion from immediately reversing the four categories, participants next completed another practice round in Block 5, in which only two categories were presented, but in reversed order compared to the first block (e.g., if prestige had been shown on the left side and dominance on the right side in Block 1, these positions were now reversed). Thereafter participants proceeded with Blocks 6 and 7. For participants for whom Blocks 3 and 4 had been congruent, the final two blocks were now incongruent; for participants for whom Blocks 3 and 4 had been incongruent, the final two blocks were now congruent. The order of congruent versus incongruent blocks as well as the placement of categories on the left versus right side of the screen was counterbalanced. Each time participants made an error they had to wait briefly before they could continue, creating an incentive to respond quickly yet accurately.

At the end of the study participants provided demographic information. After that, they were debriefed, thanked for their participation, remunerated, and dismissed.

## Results

Data preparation and calculation of IAT scores were conducted using the improved D600 scoring algorithm developed by Greenwald et al. [70]. Following this protocol, we excluded participants who responded below 300ms on more than 10% of all trials ($n = 2$), removed trials with latencies under 400ms or over 10,000ms, and added penalties for incorrect trials to the respective latencies. We subsequently computed the difference between the mean response times for norm violation/dominance and norm abidance/prestige trials (congruent blocks) and those for norm abidance/dominance and norm violation/prestige trials (incongruent blocks) by subtracting the average response time on Block 3 from the average response time on Block 6 and subtracting the average response time on Block 4 from the average response time on Block 7 (see Table 1). Each difference was divided by its pooled standard deviation as calculated based on the response times after deletion of extreme scores, but before imputation of incorrect trials. The average of these two indices constitutes the individual IAT score. A positive IAT score would therefore mean that the updated latencies (which reflect both speed and accuracy) were higher in incongruent blocks (where dominance was paired with norm abidance and prestige with norm violation) compared with congruent blocks (where dominance was paired with norm violation and prestige with norm abidance), whereas a negative score would indicate the opposite.

We found that the IAT score was positive and significant, $d = 1.03$, $t(132) = 19.44$, $p < .001$. This indicates that participants were quicker to associate norm violation with dominance and norm abidance with prestige than they were to associate norm violation with prestige and

norm abidance with dominance. This pattern suggests that our participants held implicit associations between norm violation and dominance and between norm abidance and prestige.

## Discussion

The data of this first study indicate that individuals harbor implicit associations between norm violation and dominance, and between norm abidance and prestige. This pattern of results is consistent with the relationships proposed in Hypothesis 1 (norm violations increase perceptions of dominance) and Hypothesis 2 (norm violations decrease perceptions of prestige), and it suggests that the dominance/prestige framework can be meaningfully applied to the study of social norms. Obviously, however, these data do not allow for causal conclusions concerning the relationship between norm violation and dominance and the relationship between norm abidance and prestige; the data merely reflect correlational patterns of associations. We set out to address this limitation in Study 2.

## Study 2

The aim of Study 2 was to establish causal effects of norm violation versus abidance on perceptions of dominance and prestige. To this end we created a scenario depicting an everyday situation (parking a car) in which a protagonist behaved normatively or counter-normatively, and we measured participants' perceptions of the protagonist's dominance and prestige.

## Methods

**Participants and design.** A power analysis using G*Power [69] indicated a minimum sample size of 140 participants to detect a medium-sized effect ($d$ = 0.5) using an independent-samples $t$ test with $\alpha$ = .05 and power = 0.90. We recruited a slightly larger number of participants to account for possible dropout. A total of 163 undergraduate students enrolled in the study in return for course credit. Three participants were excluded because they completed less than 75% of the questions (no forced answers were applied), and an additional 15 participants were excluded for failing an attention check (see below). The final sample consisted of 145 participants (113 women, 32 men; $M_{age}$ = 20.56, $SD_{age}$ = 4.46). Sensitivity analysis indicated that with the standard criteria of $\alpha$ = .05 (two-tailed) and power of 80% this sample size was sufficient to detect an effect size of $d$ = 0.47 or greater (i.e., a medium-sized effect).

The experiment followed a one-factor (norm violation vs. norm abidance) between-subject design and was administered online via Qualtrics. The study was approved by the Ethics Review Board of the Faculty of Social and Behavioral Sciences of the University of Amsterdam (case number: 2020-SP-12759). Data were collected between 11 November 2020 and 5 January 2021. All participants provided written informed consent.

**Procedure and materials.** Upon entering the study, participants were welcomed and instructed to immerse themselves in the following car parking scenario. Participants were randomly assigned to either the norm violation condition (in which the protagonist parked in a parking spot for disabled people) or the norm abidance condition (in which the protagonist parked in a regular parking spot).

*Imagine that you are on a road trip in another country for the first time. You just arrived in a small town in that country and dropped by a supermarket to buy something to drink. The parking lot was crowded with cars and the sun was burning. You felt very hot and thirsty and really wanted to find a spot asap so that you could buy something cold to drink. When you were looking for a place to park in the parking lot, you saw a car parking [in the parking spot for disabled persons / in a regular parking spot] in front of you. A fit-looking citizen*

*athletically stepped out of the car and left. Because you could not find a spot for your car, you decided to drive around the block in search for a spot elsewhere. When you returned, you noticed that person's car was still parked [in the spot for disabled persons / in the same parking spot], while the driver was nowhere to be seen.*

We emphasized the protagonist's physical fitness, because a pilot study had revealed that omitting this information left room for participants to infer that the protagonist was himself disabled, which would render the manipulation invalid. Adding an explicit statement about the protagonist's physical fitness ensured that their decision to park in a spot for disabled persons would be perceived as a norm violation as intended.

*Dominance and prestige.* After reading the vignette, participants indicated their perceptions of the protagonist's dominance and prestige by completing the validated dominance and prestige scales developed by Cheng and colleagues [72; peer-report versions]. The dominance scale consisted of eight items (e.g., "S/he is willing to use aggressive tactics to get his/her way," "S/he does not have a forceful or dominant personality [reverse-scored]," "Others know it is better to let him/her have his/her way," "Some people are afraid of him/her"; 1 = *not at all* to 7 = *very much*; α = .87). The prestige scale consisted of nine items (e.g., "Members of his/her group respect and admire him/her," "Members of his/her group do not want to be like him/her" [reverse-scored], "S/he is held in high esteem by members of the group," "His/her unique talents and abilities are recognized by others"; 1 = *not at all* to 7 = *very much*; α = .77). For the full scales, see Cheng et al. [72].

*Manipulation check.* To check the effectiveness of the manipulation, we adapted a manipulation check previously used by Stamkou et al. [30]. This scale consisted of three items: "To what extent do you feel that [the protagonist] broke / complied with / adhered to the rules?" (7-point scale, 1 = *not at all*, 7 = *very much*; α = .94). The second and third items were reverse-coded so that higher scores indicate greater perceived violation.

*Additional measures.* For replication purposes, we also included a validated 8-item scale developed by Anderson et al. [73] to measure participants' perceptions of the protagonist's power (e.g., "He/she can get people to listen to what s/he says"; 1 = *strongly disagree*, 7 = *strongly agree*; α = .70). We also included a number of additional measures for exploratory purposes, which did not yield insights pertinent to the present investigation. Details about these measures are provided in the Supporting Information.

*Attention check.* We included an attention check toward the end of the study. Participants were asked in which spot the protagonist mentioned in the scenario parked their car (A = *in a parking spot for disabled persons*; B = *in a regular parking spot*). Participants in the norm violation condition should answer A, and those in the norm abidance condition should answer B. Participants who answered incorrectly were excluded from the analyses (*n* = 15).

Finally, participants completed demographic questions, received a written debriefing, and were thanked and compensated for their participation.

## Results

**Manipulation check.** A significant effect of condition on the manipulation check indicated that participants in the norm-violation condition rated the protagonist's parking behavior as more transgressive (*M* = 5.83, *SD* = 1.03) than did participants in the norm-abidance condition (*M* = 2.70, *SD* = 1.66), *t*(84.00) = 12.70, *p* < .001, *d* = 2.38 (degree of freedom adjusted to account for unequal variances). We therefore conclude that the manipulation was successful.

**Dominance.** A significant effect of condition on the protagonist's perceived dominance revealed that participants in the norm-violation condition rated the protagonist as more

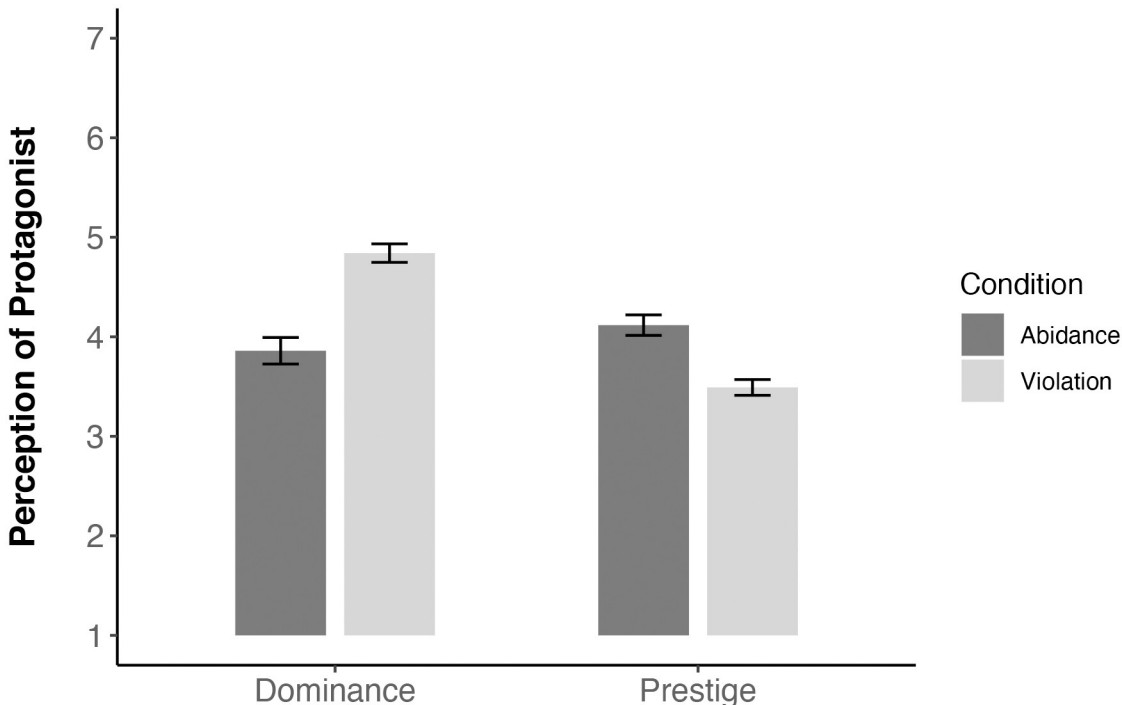

**Fig 2. Effects of norm violation versus abidance on perceived dominance and prestige (Study 2).** Dominance and prestige were rated on 7-point scales, with higher scores indicating greater endorsement. Error bars denote standard errors.

dominant ($M$ = 4.84, $SD$ = 0.87) than did participants in the norm-abidance condition ($M$ = 3.86, $SD$ = 1.01), $t$(143) = 6.22, $p$ < .001, $d$ = 1.06 (see Fig 2).

**Prestige.** A significant effect of condition on prestige showed that participants in the norm-violation condition rated the protagonist as less prestigious ($M$ = 3.49, $SD$ = 0.74) than did participants in the norm-abidance condition ($M$ = 4.12, $SD$ = 0.78), $t$(142) = -4.84, $p$ < .001, $d$ = -0.83 (see Fig 2).

**Perceived power.** Replicating previous research [24], participants perceived the norm-violating confederate as more powerful ($M$ = 4.68, $SD$ = 0.74) than the norm-abiding confederate ($M$ = 4.40, $SD$ = 0.69), $t$(140) = 2.26, $p$ = .025, $d$ = 0.39.

## Discussion

Extending the correlational findings of Study 1, Study 2 provides causal evidence that norm violators are perceived as more dominant (consistent with Hypothesis 1) and as less prestigious (consistent with Hypothesis 2) than norm abiders. Together, the results of these two studies indicate that (counter)normative behavior cues perceptions of dominance and prestige, which are known to differentially predict voluntary influence granting. In the ensuing studies we extended our investigation by examining other types of norm violations and considering their downstream consequences for influence granting.

## Study 3

The first aim of Study 3 was to replicate the effects of norm violation versus abidance on dominance and prestige and to investigate how these effects are modulated by whether local group norms and/or global community norms are violated, and what the downstream consequences

are for people's willingness to afford influence to norm violators versus norm abiders. To examine this, we created a scenario about a student graduation ceremony in which violation of or abidance by norms of the local group (the protagonist's sorority/fraternity) and norms of the global community (the protagonist's university) can be manipulated orthogonally.

On the basis of our theoretical argument that norm violations convey dominance by signaling preparedness to withstand any backlash that may result from the violation, we anticipated that violations of local group norms as well as violations of global community norms would result in heightened perceptions of dominance. With regard to prestige, the theoretical notion that prestige tracks individuals' value for the group, in particular [33], suggests that ratings of prestige respond more strongly to violations of local group norms than to violations of global community norms. The combined focus in the current study on global community norms and local group norms allowed us to examine this idea. Specifically, we tested our hypothesis that when faced with a conflict between global community norms and local group norms, protagonists amass greater prestige within their group when they choose to follow local group norms and violate global community norms than vice versa (Hypothesis 3).

The second aim of Study 3 was to conduct a first test of the consequences of local group norm and/or global community norm violations for voluntary influence granting. Specifically, we tested our hypothesis that, when local group norms and global community norms conflict, people are more willing to confer influence to individuals who violate community norms while abiding by group norms than to individuals who violate group norms while abiding by community norms (Hypothesis 4). Finally, we examined whether such preferential influence granting to community norm violators, if observed, is mediated by prestige (Hypothesis 5). We tested these hypotheses by examining people's willingness to endorse a protagonist's leadership, which served as our operationalization of influence granting.

## Methods

**Participants and design.** Power analysis using G*Power revealed that a minimum sample size of 232 participants was needed to detect a medium-sized effect ($f = .25$) with a power of .90 and $\alpha = 0.05$ when conducting ANOVA with four groups. We oversampled to account for participant dropout, which is prevalent in studies that employ online participant platforms as we did here, and because we anticipated having to filter out participants for whom the scenario may have been more difficult to relate to (see below). Based on these considerations, we recruited a total of 547 participants through CrowdFlower [74]. Because the scenario employed in this study involved a graduation ceremony of the type that is common in the US academic system, we recruited participants from the US and excluded participants who had never attended university from the analysis ($n = 99$). Additionally, participants were excluded if they did not pass an attention check ($n = 31$) or if their response patterns across all scales were suggestive of insufficient effort responding as indicated by negative person-total correlations across all items ($n = 19$), a recommended practice for online research [75]. Finally, we excluded duplicate responses (as reflected in duplicate IP addresses; $n = 24$). All in all, 374 participants were retained for analyses (240 women, 134 men; $M_{age} = 35.95$, range: 18–76). Sensitivity analysis indicated that, based on ANOVA with $\alpha = .05$ and 80% power, this sample size allowed detecting an effect size of $f = .17$ or larger (i.e., a small to medium effect).

Participants were randomly assigned to the conditions of a 2(group norm: violation vs. abidance) x 2(community norm: violation vs. abidance) between-subjects design. The study was approved by the Ethics Review Board of the Faculty of Social and Behavioral Sciences of the University of Amsterdam (case number: 2016-WOP-7376). Data were collected between 1 November 2016 and 17 November 2016. All participants provided written informed consent.

**Procedure and materials.**   In the scenario, participants read that the university where the graduation ceremony was held prescribed the color of the stole (purple or green, counterbalanced) that graduates had to wear (global community norm). Additionally, a sorority/fraternity prescribed a stole color (purple or green, counterbalanced) for graduating sorority/ fraternity members (local group norm). Depending on condition, the norms of the university and the sorority/fraternity were aligned or misaligned to enable a full factorial design in which global community norms and local group norms could be manipulated orthogonally. That is, community and group norms were aligned (i.e., both prescribed purple or green stoles) in the conditions in which the protagonist violated both norms or abided by both norms, and they were misaligned (i.e., one prescribed a green stole and the other a purple stole) in the conditions in which the protagonist abided by one norm while violating another. Thus, participants learned about a graduating fraternity/sorority member who dressed up in such a way for the ceremony that s/he either abided by norms at both levels, violated norms at both levels, abided by the group norm and violated the community norm, or violated the group norm and abided by the community norm. Participants who self-identified as female read about a sorority member and participants who self-identified as male read about a fraternity member.

After reading about the graduating sorority/fraternity member, participants completed a measure of influence granting (operationalized as leadership endorsement), rated the protagonist's dominance and prestige, and completed manipulation checks. Considering that the order in which measures are presented can influence participants' responses, and given that we already demonstrated correlational (Study 1) and causal (Study 2) associations between norm violation versus abidance and dominance versus prestige, we measured dominance and prestige after leadership endorsement to rule out that any effects observed on leadership endorsement could be due to the preceding measures of dominance and prestige.

*Influence granting*: *Intersubjective leadership endorsement*. We operationalized influence granting as the willingness to endorse the protagonist's leadership [30, 49, 76, 77], using an intersubjective perspective [78]. That is, we asked participants to what extent they thought fellow sorority/fraternity members would support the graduating member in her/his candidacy for the sorority/fraternity's alumni board. We adopted this approach to diminish participants' tendencies toward socially desirable responding, which can be a concern when asking about potentially sensitive topics [79]. Indirect questioning, such as in the intersubjective approach, is an effective strategy for circumventing social desirability bias [80]. We measured intersubjective leadership endorsement using four items that were adapted from Stamkou et al. [30] to fit the current context, asking participants to indicate how they believed fellow fraternity or sorority members of the protagonist would respond to the protagonist in the scenario: "They would vote for him/her," "They would think that s/he is a good candidate", "They would strongly support her/him," and "They would encourage others to vote for her/him" (1 = *completely disagree* to 7 = *completely agree*; α = .98).

*Dominance and prestige*. For the sake of consistency, we also measured dominance and prestige from an intersubjective perspective [78]. We used the same dominance (current α = .85) and prestige (current α = .90) scales as in Study 2 [see 72 for details], with the phrasing being adjusted in light of the intersubjective approach.

*Manipulation checks*. Finally, two manipulation checks with three analogous items each assessed perceived violations of community norms and violations of group norms ("S/he violated norms of the university [the sorority/fraternity]", "S/he behaved in line with norms of the university [the sorority/fraternity]", and "S/he behaved appropriately in the eyes of the university [the sorority/fraternity]"; 1 = *completely disagree* to 7 = *completely agree*; both αs = .91; adapted from Stamkou et al. [30]. We reverse-coded the last two items so that higher scores on these measures reflect greater perceived violation.

*Additional measures.* We included a number of additional measures for exploratory purposes. These measures did not yield insights relevant to the present investigation. Details can be found in the Supporting Information.

At the end of the study, participants provided demographic information, upon which they received a written debriefing about the purpose of the study. Finally, they were thanked for their participation and remunerated.

## Results

**Manipulation checks.** If the manipulations were successful, participants in the group norm violation conditions should report having perceived greater group norm violations than participants in the group norm abidance conditions. Furthermore, participants in the community norm violation conditions should report having perceived greater community norm violations than participants in the community norm abidance conditions. We conducted two separate 2 x 2 ANOVAs (one for each manipulation check scale) to examine whether the manipulations had these intended effects, and to gauge to what extent there was spillover between the manipulations.

A main effect of the group norm manipulation indicated that, as intended, participants in the group norm violation conditions perceived greater group norm violation ($M = 5.23$, $SD = 1.33$) than did participants in the group norm abidance conditions ($M = 2.24$, $SD = 1.13$), $F(1, 370) = 570.82$, $p < .001$, $\eta_p^2 = .61$. We also observed an unexpected main effect of the community norm manipulation on the group norm manipulation check, $F(1, 370) = 7.56$, $p = .006$, $\eta_p^2 = .02$, which was qualified by an (also unanticipated) interaction, $F(1, 370) = 13.77$, $p < .001$, $\eta_p^2 = .04$. Examination of the interaction pattern revealed that the protagonist who violated group norms was perceived as having done so to a greater extent when the group norm violation coincided with a community norm violation ($M = 5.62$, $SD = 1.26$) compared to when there was no community norm violation ($M = 4.80$, $SD = 1.29$). The protagonist who abided by the group norm was perceived as abiding by the group norm to the same extent irrespective of whether the group norm abidance coincided with a community norm violation ($M = 2.18$, $SD = 1.06$) or with community norm abidance ($M = 2.29$, $SD = 1.20$). These effects indicate that there was some spillover from the community norm manipulation onto the group norm manipulation check. However, the effect sizes of these unanticipated effects ($\eta_p^2 = .02$ and $\eta_p^2 = .04$, respectively) were negligible compared to the effect size of the intended effect ($\eta_p^2 = .61$), so we conclude that, although not perfect, the group norm manipulation was for the most part successful.

A main effect of the community norm manipulation indicated that, as intended, participants in the community norm violation conditions perceived greater community norm violation ($M = 5.15$, $SD = 1.51$) than did participants in the community norm abidance conditions ($M = 2.25$, $SD = 1.29$), $F(1, 370) = 404.43$, $p < .001$, $\eta_p^2 = .52$. There was also an unexpected main effect of the group norm manipulation on the community norm manipulation check, such that the protagonist who had violated the group norm was perceived as also having violated the community norm to a somewhat greater extent ($M = 3.98$, $SD = 1.96$) than the protagonist who had abided by the group norm ($M = 3.43$, $SD = 2.04$), $F(1, 370) = 10.10$, $p = .002$, $\eta_p^2 = .03$. The interaction between group and community norm violation versus abidance was non-significant, $F(1, 370) = 1.84$, $p = .176$, $\eta_p^2 = .01$. These data indicate that there was some spillover from the group norm manipulation onto the community norm manipulation check. However, given the large effect size of the intended effect ($\eta_p^2 = .52$) and the very small effect size of the unintended side-effect ($\eta_p^2 = .03$), we conclude that the community norm manipulation largely worked as intended.

**Table 2. Effects of violation of versus abidance by group and community norms on dominance, prestige, and intersubjective leadership endorsement (Study 3).** Dominance, prestige, and leadership endorsement were rated on 7-point scales, with higher scores reflecting greater endorsement. Means within a row with a different subscript differ at $p < .05$ according to Bonferroni-Holm-corrected pairwise comparisons.

| Condition | | | | |
|---|---|---|---|---|
| **Group Norm** | **Abidance** | | **Violation** | |
| **Community Norm** | **Abidance** | **Violation** | **Abidance** | **Violation** |
| Measure | | | | |
| Dominance | 3.74 (0.78) a | 4.44 (1.15) b | 3.90 (0.91) a | 4.71 (0.90) b |
| Prestige | 4.92 (0.84) a | 5.02 (0.87) a | 4.17 (0.99) b | 3.99 (0.88) b |
| Intersubjective Leadership Endorsement | 5.06 (1.12) a | 5.61 (1.02) b | 3.40 (1.44) c | 2.98 (1.48) d |

**Dominance.** We predicted that people who violate norms appear more dominant than people who abide by norms (Hypothesis 1), and we expected this pattern to hold for global community norm violations as well as local group norm violations. The most direct test of this hypothesis is by means of one-way ANOVA with follow-up planned contrasts. If Hypothesis 1 held true, we would expect to see higher dominance ratings of a protagonist who violated global community norms and/or local group norms than of a protagonist who abided by both norms. One-way ANOVA revealed significant differences in dominance ratings among the four conditions, $F(3, 370) = 22.28$, $p < .001$, $\eta_p^2 = .15$, indicating that our manipulation had an impact on participants' ratings of the protagonist's dominance (see Table 2 for means and standard deviations). More important, and as predicted, contrast analysis revealed that dominance ratings were significantly higher when the protagonist violated one or more norms (the global community norm, the local group norm, or both; $M_{pooled} = 4.36$, $SD_{pooled} = 1.05$) than when the protagonist abided by both norms ($M = 3.74$, $SD = 0.78$), $t(370) = 5.53$, $p < .001$, $d = 0.63$. These results are in line with Hypothesis 1.

**Prestige.** We predicted that, when faced with a conflict between local group norms and global community norms, individuals gain more prestige within their group when they follow local group norms and violate global community norms than when they follow global community norms and violate local group norms (Hypothesis 3). Support for this hypothesis would entail significantly higher ratings of prestige of a protagonist who violates community norms while abiding by conflicting group norms than of a protagonist who exhibits the opposite pattern. One-way ANOVA revealed significant differences in prestige ratings among the four conditions, $F(3, 370) = 31.68$, $p < .001$, $\eta_p^2 = .20$, indicating that the manipulations affected participants' ratings of the protagonist's prestige (see Table 2 for $M$s and $SD$s). More important, and in line with our hypothesis, contrast analysis showed that more prestige was attributed to a protagonist who violated community norms while abiding by (conflicting) group norms ($M = 5.02$, $SD = 0.87$) than to a protagonist who violated group norms while abiding by (conflicting) community norms ($M = 4.17$, $SD = 0.99$), $t(370) = 6.41$, $p < .001$, $d = 0.91$.

*Influence granting*: *Intersubjective leadership endorsement*. We predicted that, when local group norms and global community norms conflict, protagonists who violate community norms while abiding by group norms receive greater endorsement as leaders than those who violate group norms while abiding by community norms (Hypothesis 4). One-way ANOVA revealed significant differences in leadership endorsement among the four conditions, $F(3, 370) = 92.24$, $p < .001$, $\eta_p^2 = .43$ (see Table 2 for $M$s and $SD$s). More specifically, and consistent with Hypothesis 4, contrast analysis showed that leadership endorsement was higher when the protagonist violated community norms while abiding by (conflicting) group norms ($M = 5.61$, $SD = 1.02$) than when the protagonist violated group norms while abiding by (conflicting) community norms ($M = 3.40$, $SD = 1.44$), $t(370) = 11.60$, $p < .001$, $d = 1.77$.

*Additional analyses*. We also analyzed effects on dominance, prestige, and influence granting using 2 x 2 ANOVAs in which the effects of abidance by versus violation of community norms and abidance by versus violation of group norms can be tested separately and in interaction. These analyses yielded similar conclusions and are reported in the Supporting Information.

*Mediation*. Thus far we have established that a protagonist who violated community norms while abiding by (conflicting) group norms was perceived as more prestigious (in line with Hypothesis 3) and was more likely to be endorsed as a leader (in line with Hypothesis 4) compared to a protagonist who violated group norms while abiding by (conflicting) community norms. The final step in our analysis is to test whether the preferential leadership endorsement of community norm violators is mediated by prestige, as predicted under Hypothesis 5. Although mediation analysis is inherently correlational and alternative models are therefore conceivable (e.g., influence granting may drive perceptions of prestige), we deem the current sequence of variables more plausible as prestige is widely understood to be a precursor to influence granting [32–34]. To control for possible influences of dominance perceptions, we conducted a multiple mediation analysis with the contrast between the community norm violator and the group norm violator as the predictor, dominance and prestige as simultaneous candidate mediators, and leadership endorsement as the dependent variable. Bootstrapped confidence intervals based on 10,000 bootstrap samples yielded a significant indirect effect through prestige, $b = 0.79$, 95% CI [0.53, 1.06], and no indirect effect through dominance, $b = 0.07$, 95% CI [-0.00, 0.17]. Path coefficients are presented in Fig 3. Repeating the analysis without controlling for dominance produced a similar indirect effect via prestige, $b = 0.77$, 95% CI [0.52, 1.04]. Even though these results do not provide causal evidence for the role of prestige, they are consistent with our theoretical argument that prestige mediates the preferential

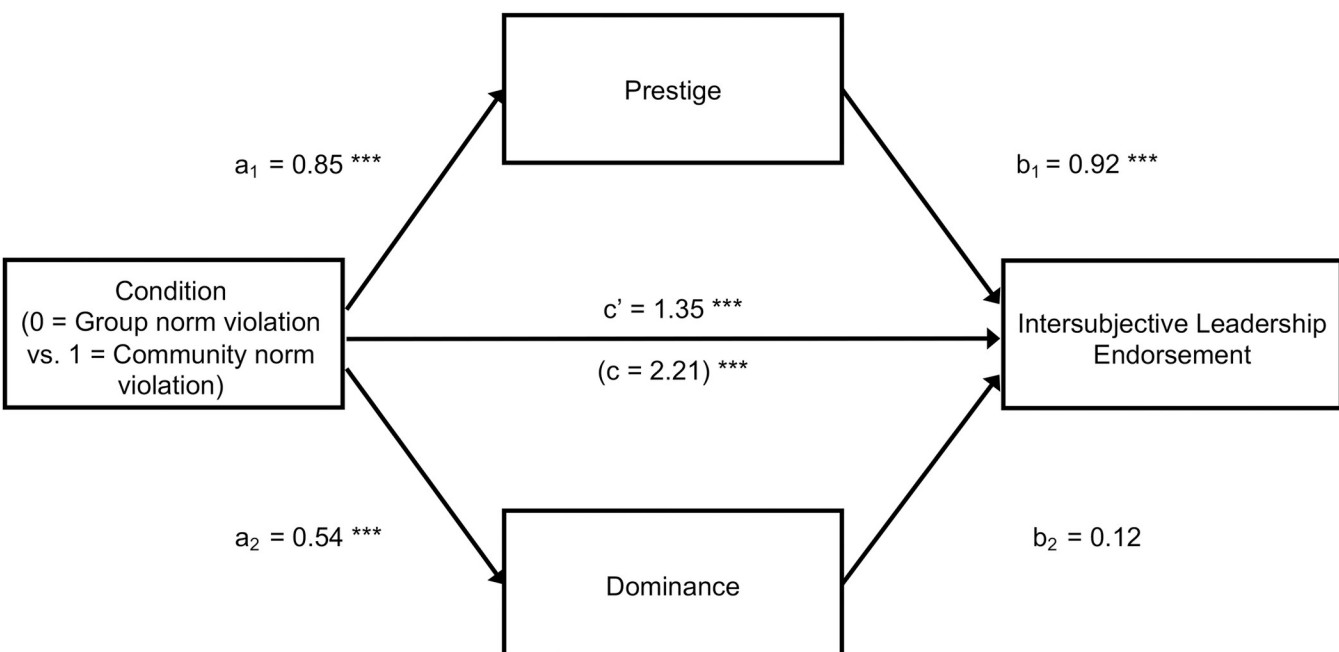

**Fig 3. Multiple mediation model testing the effect of type of norm violation on intersubjective leadership endorsement via prestige and dominance (Study 3).** In the community norm violation condition, a protagonist violated the global community norm and abided by the local group norm; in the group norm violation condition, the protagonist violated the local group norm and abided by the global community norm. Unstandardized regression coefficients are reported.

endorsement of community norm violators as leaders in situations where group norms and community norms conflict (i.e., Hypothesis 5).

## Discussion

Study 3 replicates and extends our previous findings in several ways. First, we obtained additional evidence that norm violations fuel perceptions of dominance (supporting Hypothesis 1), and this held for global community norm violations as well as local group norm violations. Second, we obtained first evidence that, when group norms and community norms conflict, individuals who violate community norms while abiding by group norms are perceived as more prestigious by their fellow group members than individuals who violate group norms while abiding by community norms (supporting Hypothesis 3). Third, we obtained support for our prediction that, when group and community norms conflict, people are more willing to grant influence to individuals who violate community norms while abiding by group norms than to individuals who do the opposite, as reflected in greater leadership endorsement ratings of community norm violators than of group norm violators (supporting Hypothesis 4). Interestingly, as seen in Table 2, the combination of local group norm abidance and global community norm violation elicited greater leadership endorsement than any other combination, including overall abidance. We speculate that this may be because, unlike individuals who follow both group and community norms, community norm violators signal they are willing to incur a risk in order to honor the group's norms. This may increase their perceived loyalty and commitment to the group [58], which may make them even more attractive as leaders in the eyes of fellow group members. Finally, we found that preferences for community norm violators as leaders were mediated by prestige (supporting Hypothesis 5).

Two possible limitations of this study merit discussion. First, our manipulation checks indicated that the effects of global community norm abidance versus violation and local group norm abidance versus violation were not fully orthogonal as there was some spillover between the manipulations. The sizes of the unintended effects are negligible compared to those of the intended effects, but it is possible that the results we obtained are attenuated somewhat by this imperfect orthogonality.

Second, we employed intersubjective measures of dominance, prestige, and leadership endorsement in this study to alleviate possible concerns about socially desirable responding [80]. This approach was successful in that we obtained meaningful and predicted differences between the experimental conditions, implying that the predicted patterns were not obfuscated by floor effects that might have resulted from social desirability concerns. A downside of the intersubjective approach is that it taps people's naïve understanding of how others respond to norm violators rather than their own responses. We deemed this sacrifice warranted against the background of the results from our previous studies, which had already established the hypothesized links between norm violation versus abidance and dominance versus prestige in the form of generic associations (Study 1) a well as impressions from a first-person perspective (Study 2). The consistency of the Study 3 results pertaining to dominance and prestige with the results from Studies 1 and 2 enhances confidence that the approach we took in Study 3 is valid. Indeed, Chiu and colleagues [78] have argued that intersubjective perceptions are important predictors of people's own responses–"at times, more . . . than . . . personal values and beliefs" (p. 483). Clearly, however, Studies 1 and 2 do not speak to the effects of norm violations on influence granting, and therefore more evidence is needed before we can draw firm conclusions about the consequences of different types of norm violations for people's ability to attain influence in groups. Thus, a central goal of our final two studies was to examine whether such evidence can be obtained using alternative methodological approaches.

## Study 4

Study 3 provided initial evidence that, when local group norms and global community norms conflict, people are more willing to grant influence (via leadership endorsement) to others who violate community norms while abiding by group norms rather than vice versa. In Study 4 we sought to extend this finding by examining whether individuals who more frequently violate global community norms as opposed to local group norms are seen as more popular, which would put them in a favorable position to exert influence in the group. If individuals who prioritize abidance by local group norms over abidance by global community norms are perceived in ways that render them more likely to attain positions of influence, as Study 3 suggests, then we would expect that people who are popular and therefore have the potential to be influential are more likely to be those who preferentially violate community rather than group norms (cf. Hypothesis 4). We investigated this possibility in a school context, within which popular students can be considered leaders in an informal hierarchy [81]. The school context is a classic example of a nested social setting, in which groups of students (i.e., classes or cliques) are nested within a larger community (i.e., the school). In this context, students may behave more or less in line with the norms of their peers (representing the local group) and/or with the norms of the teachers (representing the school). Situating our study in this context enabled us to examine associations between different constellations of (counter)normative behavior and potential for influence in the form of popularity. As such, this study also extends our investigation to informal (rather than formal) positions of influence [82].

### Methods

**Participants and design.** Power analysis using G*Power recommended a minimum sample size of 30 to detect a medium-sized effect of $f = 0.25$ in a repeated-measures ANOVA involving within-participant comparisons across 4 measurements with $\alpha = 0.05$ and power = .90. We oversampled to account for possible participant dropout, recruiting 56 bachelor students from the University of Amsterdam in exchange for course credits (51 women, 5 men; $M_{age} = 20.05$, range: 18–30). One participant was excluded due to a failed attention check. Sensitivity analysis indicated that, based on a repeated-measures ANOVA with $\alpha = .05$ and 80% power, the final sample size was sufficient to detect an effect size of $f = .16$ or larger (i.e., a small to medium effect).

The study was approved by the Ethics Review Board of the Faculty of Social and Behavioral Sciences of the University of Amsterdam (case number: 2016-WOP-7439). Data were collected between 22 November 2016 and 7 December 2016. All participants provided written informed consent.

**Procedure and materials.** Participants were instructed to think back to their time in high school. Once they had activated their memories of their school, participants were asked to describe two students from their school: a popular student and a student who was neither popular nor unpopular (henceforth: "control student"). The order in which participants were prompted to reflect on the popular student and the control student was randomized.

To ensure that participants had two actual people in mind, they were asked to describe each student in some detail (e.g., gender, hair color, posture, personality). Next, they indicated how frequently each student would display behaviors that were disapproved of by the student's peers and how frequently the student would display behaviors that were disapproved of by the student's teachers, using a 100-point slider scale (1 = *never* to 100 = *always*). The rated frequency of the student's violation of peer norms served as our measure of local group norm violation; the rated frequency of the student's violation of teachers' norms served as our measure of global community norm violation.

As an alternative operationalization of the violation of community norms, we also asked participants to rate the frequency with which the popular student and the control student would violate norms that were endorsed by students outside of the peer group, using the same 100-point slider scale. Analyses involving this measure instead of the measure tapping violation of teachers' norms yielded similar results. Details are presented in the Supporting Information. In addition, we included a number of exploratory measures, which did not yield pertinent insights. Details about these measures can also be found in the Supporting Information.

At the end of the study, participants provided demographic information. After that, they received a written debriefing. Finally, participants were thanked and compensated.

## Results

A repeated-measures ANOVA with target (popular vs. control student) and norms (teachers' norms vs. peers' norms) as within-participants factors revealed significant main effects of target, $F(1, 54) = 16.34$, $p < .001$, $\eta^2 = 0.06$, and norms, $F(1, 54) = 30.72$, $p < .001$, $\eta^2 = 0.09$, as well as a significant interaction between target and norms, $F(1, 54) = 37.49$, $p < .001$, $\eta^2 = 0.08$. Follow-up analyses revealed no significant difference between popular students ($M = 18.73$, $SD = 16.17$) and control students ($M = 20.47$, $SD = 22.20$) in the frequency with which they were recalled to violate the norms of their peers (local group norms), $t(54) = -0.59$, $p = .557$, $d = 0.08$. There was, however, a significant difference in the degree to which the two types of students were recalled as violating the norms of their teachers (global community norms), such that popular students were remembered as having violated teachers' norms more frequently ($M = 46.00$, $SD = 26.18$) than control students ($M = 21.51$, $SD = 23.36$), $t(54) = 6.07$, $p < .001$, $d = 0.82$. Moreover, the popular student was recalled as having more frequently violated teacher norms than peer norms ($M = 46.00$ vs. $M = 18.73$, respectively), $t(54) = 7.61$, $p < .001$, $d = 1.08$. No such effect was observed for the control student ($M = 21.51$ vs. $M = 20.47$, respectively), $t(54) = 0.34$, $p = .736$, $d = 0.05$. These results are graphically depicted in Fig 4.

## Discussion

The results of Study 4 further support the idea that responses to norm violations are not monolithic but depend on the type of norms being violated. Specifically, high school students who were seen as popular and therefore presumably had greater influence among their fellow students were recalled as having more frequently violated norms endorsed by their teachers (i.e., global community norms) compared to students who were not particularly popular and therefore presumably had less influence among their fellow students. Popular students were not recalled as more frequently violating norms endorsed by their fellow students (local group norms).

Although in line with our predictions, the results of this study do not allow for causal conclusions. It is conceivable that students who enjoy greater popularity and influence among their peers experience more leeway to violate community norms as they feel they are backed up by fellow students. Indeed, previous research has documented that people who feel more powerful are more likely to violate norms than those who feel less powerful [83, 84]. This reverse effect cannot be ruled out on the basis of the current data, and in fact the two effects may work in tandem to create the observed associations [24]. Still, the pattern observed in Study 4 is fully consistent with the causal effects of community versus group norm violations on leadership endorsement observed in Study 3, which enhances our confidence in the conclusion that people are more willing to grant influence to individuals who violate community norms while abiding by group norms than to individuals who violate group norms while abiding by community norms.

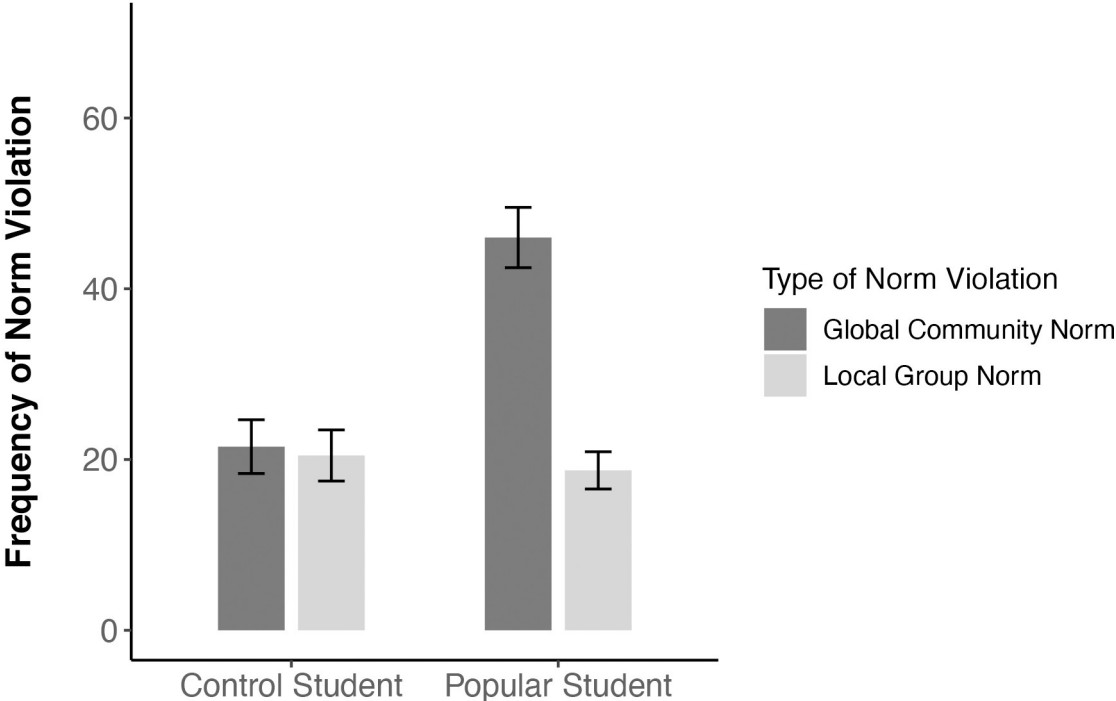

**Fig 4. Recalled frequency of global community norm violations and local group norm violations of popular students and control students (Study 4).** The recalled frequency of the different types of norm violations was rated on a 100-point scale, with higher scores indicating higher frequency. Error bars denote standard errors.

Another limitation of Study 4 is that the data provide no direct insight into whether local group norms represented by students and global community norms represented by teachers were at odds. We surmise that the likelihood of this having been the case for most participants is relatively high in this context. Indeed, the pattern we observed–that popular students more frequently violated global rather than local norms–is compatible with the possibility that these students violated global norms in order to be able to follow local norms. Clearly, however, direct evidence is lacking. In Study 5 we revert to an experimental approach to remedy these limitations and further substantiate and extend our conclusions.

## Study 5

The results of Studies 3 and 4 are consistent with the idea that individuals who violate global community norms while abiding by local group norms are perceived as prestigious by fellow group members and as fit for influential positions within the group. In our final study we examined whether local group norms and global community norms must conflict in order for these effects to emerge. Our theoretical argument suggests that individuals who violate community norms while abiding by group norms are deemed worthy of influence by fellow group members because they apparently prioritize local group norms over global community norms, which demonstrates their allegiance to the group. Given that norm violations are generally associated with lower prestige, however, as demonstrated in Studies 1 and 2, community norm violators should not be perceived as prestigious and worthy of influence when there was no need for them to violate a norm because group and community norms did not conflict and both could therefore be respected at the same time. Thus, in Study 5 we tested our hypotheses that community norm violators are perceived as more prestigious by fellow group members when local group norms and global community norms conflict–that is, when violators rebel

with a cause–than when they do not conflict (Hypothesis 6), that community norm violators are more likely to be granted influence when group and community norms conflict than when they do not conflict (Hypothesis 7), and that this greater willingness to grant influence to community norm violators under conditions of conflicting norms is mediated by prestige (Hypothesis 8).

In addition to testing these additional hypotheses, we incorporated a more direct measure of influence granting. In Study 3 we employed an intersubjective measure of leadership endorsement. Whereas this approach allowed us to circumvent possible response tendencies associated with social desirability concerns [80], intersubjective data do not directly speak to people's own willingness to grant influence to norm violators. Even though the intersubjective approach has been proven valid and superior to subjectivist measures in certain contexts (e.g., when examining sensitive issues; [78]), we deemed it important to complement the intersubjective data with more direct, behavioral data. To this end, we included a behavioral measure of influence granting in Study 5: assignment of leadership tasks.

## Methods

**Participants and design.** Power analysis using G*Power recommended a minimum sample size of 140 participants to detect a medium-sized effect ($d = 0.5$) in an independent-samples $t$ test with $\alpha = 0.05$ and power = .90. As in the previous studies, we oversampled to compensate for participant dropout, which is common in studies using the online crowdsourcing platform Amazon Mechanical Turk [85] that we used in this study. We obtained a total of 263 responses. Following preregistered exclusion criteria, we excluded duplicate responses ($n = 5$), participants who failed comprehension checks ($n = 41$), and participants whose response patterns across all scales were suggestive of insufficient effort responding ($n = 35$). This resulted in a total usable sample of 182 participants (94 women, 86 men, 2 non-binary; $M_{age} = 36.36$, range: 19–69). Sensitivity analysis indicated that, given the obtained sample size, an independent-samples $t$ test with $\alpha = .05$ (two-tailed) and 80% power could detect an effect size of $d = 0.37$ or greater (i.e., a small to medium effect).

Participants were randomly assigned to one of two conditions, in which a protagonist violated a global community norm under conditions where the global community either conflicted with a local group norm or did not conflict with the local group norm. The study was approved by the Ethics Review Board of the Faculty of Social and Behavioral Sciences of the University of Amsterdam (case number: 2017-WOP-8443) and pre-registered on OSF (https://osf.io/mndxy/?view_only=0030a11b68a04c88895b15bb14ea1b33). Specifically, the effect of our experimental manipulation on the intersubjective measure of influence granting was pre-registered as a confirmatory hypothesis (cf. Hypothesis 7); the behavioral measure of influence granting was pre-registered as exploratory. Data were collected between 1 November 2017 and 2 November 2017. All participants provided written informed consent.

**Procedure and materials.** Participants read a scenario about a telecommunications company, Mobile2, in which a team, Sim2, specialized on the Sim-only market. The norms in question pertained to the printing of logos on reports. Both the company and the team had a norm dictating that their respective logos be printed on reports. In both versions of the scenario, the protagonist decided not to print the Mobile2 logo on a report, whereas the Sim2 logo was printed. Thus, in both conditions a global (company) norm violation took place. What was varied between conditions is whether or not it was possible for employees to include multiple logos in their reports, which constituted our manipulation of norm conflict.

In the conflicting norms condition, participants learned that only one logo could be included in a report, meaning that employees had to decide whether to include the Mobile2

logo or the Sim2 logo. In this condition, employees who wanted to abide by the local (team) norm (i.e., including the Sim2 logo in their report) therefore had to violate the global (company) norm by omitting the Mobile2 logo from their report. Upon reading these instructions, participants were shown the cover page of a report that was sent around by an employee of Mobile2 working in the Sim2 team, which conspicuously included only Sim2's logo on it.

In the non-conflicting norms condition, participants read that multiple logos could be included in a report, so that employees could in principle include both the Mobile2 logo and the Sim2 logo. In this condition, employees who wanted to abide by the local (team) norm (i.e., including the Sim2 logo) therefore did not have to violate the global (company) norm (i.e., including the Mobile2 logo), because they could include both logos in their report. After reading this information, participants in this condition, too, were shown the cover page of a report issued by a Sim2 employee which only featured Sim2's logo.

Thus, in both conditions, the protagonist violated a company norm by omitting the Mobile2 logo from the report. The critical difference between the conditions was whether this omission was necessary in order to be able to abide by the local group norm of including the Sim2 logo. In the conflicting norms condition, including the Sim2 logo necessitated omitting the Mobile2 logo, making the global norm violator a rebel with a cause. In the non-conflicting norms condition, including the Sim2 logo did not force the employee to leave out the Mobile2 logo (because both logos could be included), making the global norm violator a rebel without a cause.

After reading these instructions and reviewing the cover page featuring the Sim2 logo, participants completed the dependent variables.

*Dominance and prestige*. Dominance and prestige were measured using the same scales as in Study 3 (see Cheng et al., 2010, for details). The reliabilities of both scales in the current sample were good (dominance: $\alpha = .82$; prestige: $\alpha = .94$).

*Influence granting*: *Intersubjective leadership endorsement*. We used the same intersubjective leadership endorsement scale as in Study 3 (current $\alpha = 96$).

*Influence granting*: *Assignment of leadership tasks*. We complemented the intersubjective leadership endorsement scale with a behavioral measure of influence granting. For this measure, participants read additional information about Sim2 indicating that the team was short on employees and that everyone in the team needed to pick up additional tasks. We identified twenty tasks that had been associated with leader versus subordinate roles in previous research [see 86], and conducted a pilot study to test to what extent the various tasks were indeed seen as belonging to leaders versus subordinates. Sixty-five undergraduate students (48 women, 15 men, 2 non-disclosed, $M_{\text{age}} = 22.03$, range: 19–36) rated each of the twenty tasks on two 100-mm visual analogue scales ranging from 1 = *subordinate task* to 100 = *leader task* and 1 = *low-status task* to 100 = *high-status task*.

Based on the pilot data, we selected twelve tasks that were most clearly associated with leader versus follower roles, respectively, for use in the main study. Specifically, we included the following six leader tasks: Working on articulating a new vision/goal for the department, being a guest on a television show to talk about the company's successful new product, publicly accepting an award given by the organization to the team, delivering a speech to honor a colleague's achievements, having dinner with potential new clients/customers, and presenting positive annual figures to the other departments in the company. In addition, we included the following six subordinate tasks: Sorting employee badges according to last name, handing out drinks during an office excursion, organizing catering for a staff party, collecting receipts from a conference for reimbursement, offering coffee at a business conference stand, and deciding which brand of tea will be purchased in the company.

The twelve tasks were presented in a randomized list, from which participants were instructed to select four tasks that HR should assign to the employee whom they had

previously read about; the remaining tasks would be completed by other members of the Sim2 team. Thus, the number of selected leader tasks to be assigned to the employee could range from zero to four, with higher numbers indicating greater influence granting [86].

*Manipulation check.* We measured participants' perceptions of the protagonist's norm violation using the same three items as in Study 3, which were adapted to fit the current context: "He violated norms of his company Mobile2", "He behaved in line with norms of his company Mobile2", and "He behaved appropriately in the eyes of his company Mobile2" (1 = *completely disagree* to 7 = *completely agree*; α = .95). We reverse-coded the last two items so that higher scores reflect greater perceived violation.

*Additional measures.* At the end of the study we included a number of exploratory measures, which did not yield pertinent insights. Details about these measures can be found in the Supporting Information.

Due to a programming oversight, the forced-response option was left off in this study. This means that participants could continue without answering all questions on a page. As a result, the degrees of freedom in the analyses reported below differ somewhat across analyses as we had more observations for some measures than for others. Listwise exclusion of participants who produced incomplete responses yielded similar results and conclusions.

## Results

**Manipulation check.**   Per our pre-registration, we excluded participants who failed factual comprehension checks asking (1) which logos Mobile2 and Sim2 wanted to be included in reports and (2) whether only one or multiple logos could be included. Thus, all participants who were retained for analysis had properly understood whether the norms of the company did or did not conflict with the norms of the team.

We did not anticipate a significant difference between conditions in how transgressive participants perceived the norm violator to be, because participants in both conditions were confronted with a person who violated a company norm. However, we deemed it useful to check this, as the presence of a significant difference could render any findings more difficult to interpret. An independent-samples $t$ test produced no evidence that participants perceived the behavior of the protagonist in the conflicting norms condition ($M = 4.83$, $SD = 1.88$) as more or less transgressive than the behavior of the protagonist in the non-conflicting norms condition ($M = 4.75$, $SD = 1.72$), $t(170) = 0.29$, $p = .770$, $d = 0.04$. Moreover, one-sample $t$ tests indicated that participants in both conditions scored significantly above the mid-point of the scale (conflicting norm condition: $t = 4.24$, $p < .001$; non-conflicting norm condition: $t = 3.86$, $p < .001$), indicating that they perceived the protagonist in both conditions as violating company norms, as intended.

**Dominance and prestige.**   There were no differences between the conditions in terms of perceptions of dominance (conflicting norms: $M = 3.91$, $SD = 1.07$; non-conflicting norms: $M = 3.73$, $SD = 0.94$), $t(174) = 1.17$, $p = .244$, $d = 0.18$. For prestige, we hypothesized that community norm violators are perceived as more prestigious in situations in which local group norms conflict with global community norms than in situations in which norms do not conflict (Hypothesis 6). Consistent with this hypothesis, the norm-violating employee was rated as significantly more prestigious in the condition in which community norms conflicted with group norms ($M = 5.18$, $SD = 1.05$) than in the condition in which the norms did not conflict ($M = 4.87$, $SD = 0.90$), $t(174) = 2.07$, $p = .040$, $d = 0.32$. These patterns are visualized in Fig 5.

**Influence granting: Intersubjective leadership endorsement.**   We hypothesized that community norm violators are granted more influence in situations in which local group norms and global community norms conflict than in situations in which they do not

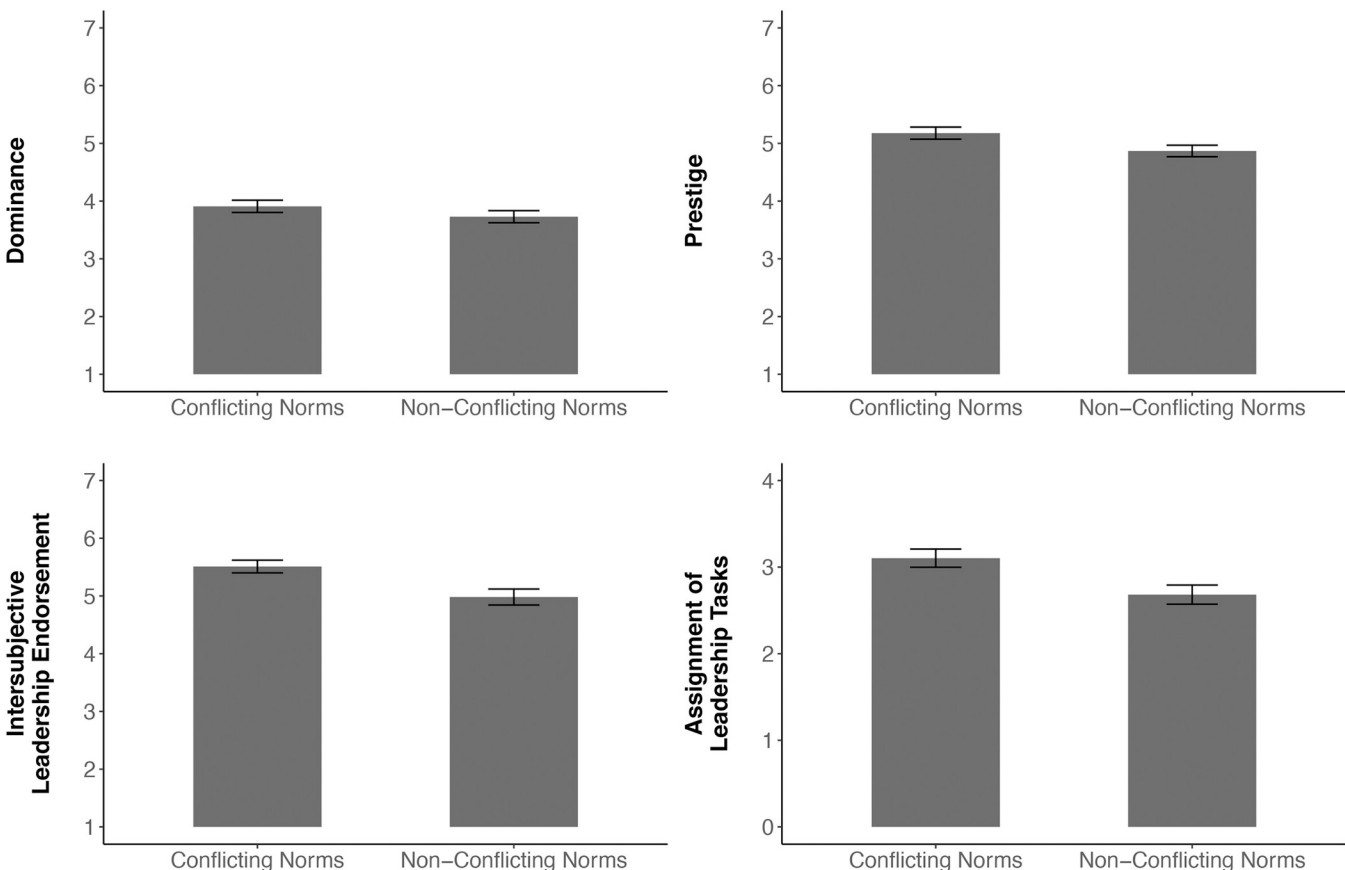

**Fig 5. Effects of community norm violation on dominance, prestige, intersubjective leadership endorsement, and assignment of leadership tasks as a function of whether or not local group norms conflict with global community norms (Study 5).** Dominance, prestige, and intersubjective leadership endorsement were rated on 7-point scales; responses on the behavioral measure of assignment of leadership tasks could range from 0 to 4 leadership tasks assigned. Error bars denote standard errors.

(Hypothesis 7). In keeping with this hypothesis, leadership endorsement ratings vis-à-vis the norm-violating employee were significantly higher in the condition in which global community norms conflicted with local group norms ($M = 5.51$, $SD = 1.11$) than in the condition in which they did not conflict ($M = 4.98$, $SD = 1.25$), $t(180) = 3.02$, $p = .003$, $d = 0.45$ (see Fig 5).

**Influence granting: Assignment of leadership tasks.** Preliminary analyses revealed that scores on the behavioral measure of leadership task assignment (which could range from 0 to 4 leadership tasks assigned) were not normally distributed, indicating that differences between conditions should be analyzed using a non-parametric test. A Wilcoxon rank-sum test showed that participants assigned significantly more leadership tasks to the norm-violating employee when group and community norms conflicted ($M = 3.10$, $SD = 1.06$) than when they did not ($M = 2.68$, $SD = 1.00$), $W = 3597$, $p = .005$, $r = .23$. (A regular independent-samples $t$ test, which is less suitable given the violation of normality, yielded a compatible result, $t[151] = 2.51$, $p = .013$, $d = 0.41$) (see Fig 5).

**Mediation.** Thus far we have established that the community norm violator was perceived as more prestigious and was granted more influence (as operationalized via intersubjective leadership endorsement and assignment of leadership tasks) when local group norms conflicted with global community norms than when they did not. The final step in our analysis is to test whether the greater influence granting to the community norm violator under

conditions of conflicting rather than non-conflicting norms is mediated by prestige, as predicted under Hypothesis 8. Although we acknowledge that alternative mediational sequences are possible, as argued in Study 3, we deem the current model most plausible in light of extensive evidence that prestige precedes influence granting [32–34]. As in Study 3, to control for possible influences of perceived dominance, we conducted a multiple mediation analysis involving both prestige and dominance as candidate mediators. In line with Hypothesis 8, bootstrapped confidence intervals (based on 10,000 bootstrap samples) revealed a significant indirect effect on the intersubjective leadership endorsement measure through prestige, $b = 0.26$, 95% CI [0.02, 0.52], and no indirect effect through dominance, $b = 0.01$, 95% CI [-0.02, 0.05]. Path coefficients are presented in Fig 6. Running the analysis without dominance in the model yielded a similar mediation via prestige, $b = 0.25$, 95% CI [0.10, 0.47].

For the behavioral measure of influence granting (i.e., the assignment of leadership tasks), we had fewer observations in the second part of our mediation model (from prestige to assignment of leadership tasks) due to missing values on the dependent variable. Calculating confidence intervals for the indirect effect requires complete data on the path from condition to mediator as well as the path from mediator to dependent variable [87]. We therefore imputed missing values on the dependent variable [88]. We used three different imputation methods (expansion, stochastic imputation, and predictive mean matching), which yielded very similar solutions and results. Here we report the results of the expansion method (a description of this approach and the results of the other approaches are provided in the Supporting Information). As before, we conducted a multiple-mediation analysis to control for possible influences of perceived dominance. This analysis (based on 10,000 bootstrap samples) revealed that prestige mediated the relationship between condition and the assignment of leadership tasks, $b = 0.12$, 95% CI [0.01, 0.26], whereas dominance did not, $b = 0.02$, 95% CI [-0.02, 0.10]–see Fig 6 for path coefficients. Running the analysis without dominance in the model yielded a similar mediation through prestige, $b = 0.12$, 95% CI [0.01, 0.25]. Although the correlational nature of mediation analysis does not allow ruling out alternative models, the results pertaining to both the intersubjective leadership endorsement measure and the behavioral measure of assignment of leadership tasks are consistent with our theoretical argument that preferential influence granting to community violators under conditions where local group norms conflict with global community norms are driven by perceptions of prestige (per Hypothesis 8).

## Discussion

The results of Study 5 indicate that community norm violators are perceived as more prestigious and are therefore more likely to be granted influence when community norms conflict with group norms–in which case the community norm violator can be seen as a rebel with a cause–than when norms do not conflict. Of note, we obtained similar effects on the intersubjective measure of leadership endorsement previously employed in Study 3 as on the behavioral measure involving the assignment of leadership tasks that we added in this study, enhancing confidence in the robustness of the findings.

## General discussion

We conducted five studies to enhance understanding of whether, when, and how norm violators gain influence in groups. Integrating insights from the dominance/prestige framework of social rank with the notion that norms are contextualized in nested social systems, we proposed that the degree of influence granted to norm violators depends on whether they violate global community norms and/or local group norms, because different constellations of norm violation versus abidance have differential consequences for actors' perceived dominance and

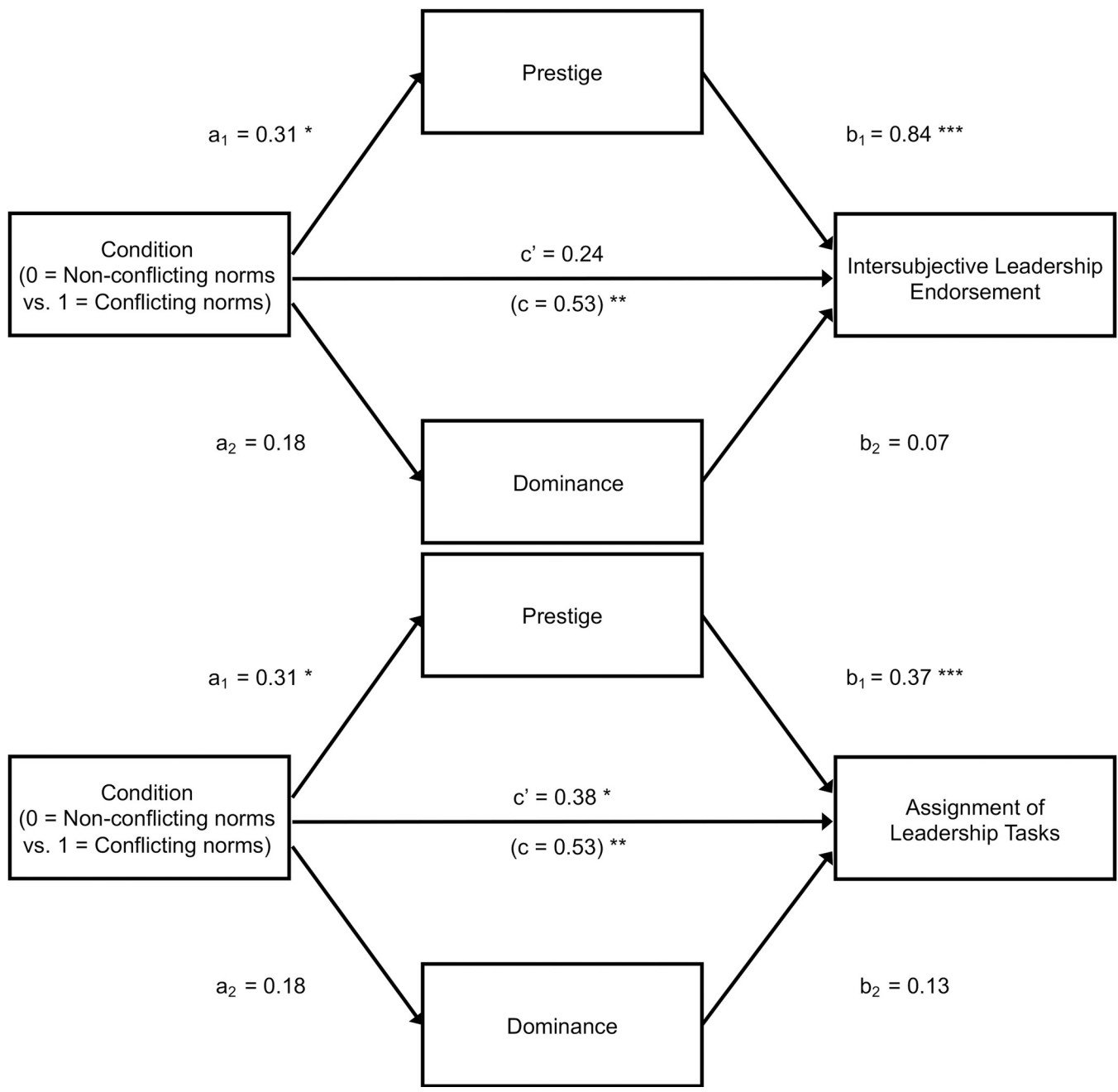

**Fig 6.** Multiple mediation models testing the effect of conflict between local group norms and global community norms on intersubjective leadership endorsement (Panel A) and assignment of leadership tasks (Panel B) via prestige and dominance (Study 5). In the conflicting norms condition, the local group norm conflicted with the global community norm; in the non-conflicting norms condition the two norms did not conflict. In both conditions the protagonist violated the community norm. Unstandardized regression coefficients are reported.

prestige. Using an implicit association test, Study 1 revealed that people implicitly associate norm violations with dominance, and norm abidance with prestige. Study 2 provided complementary causal evidence that norm violations (in this case faulty parking) can increase perceptions of dominance and decrease perceptions of prestige compared to normative behavior (regular parking). Extending these insights by comparing violations of global community norms versus local group norms, Study 3 showed that individuals who violated community

norms while abiding by (conflicting) group norms gained prestige in the eyes of fellow group members, which in turn fueled influence granting as measured through intersubjective leadership endorsement. Study 4 showed that students who were popular (and therefore presumably influential) in high school were recalled as preferentially violating the norms of teachers as opposed to those of fellow students in their peer group. Finally, Study 5 demonstrated that community norm violators were conferred more prestige and influence (as measured through intersubjective leadership endorsement as well as behavioral assignment of leadership tasks) when community norms conflicted (rather than did not conflict) with group norms, in which case community norm violators can be seen as rebels with a cause. Below we discuss theoretical and practical implications of these findings, consider strengths and limitations of our approach, and provide suggestions for future research.

## Theoretical and practical implications

Previous research on social responses to norm violators has yielded divergent patterns of results. Some studies suggested that norm violators elicit unfavorable perceptions and backlash from others that curtail their influence in groups [e.g., 7, 8, 10, 14, 15, 17, 19, 20], whereas other studies suggested that norm violators elicit favorable perceptions and supportive responses from others that enhance their influence [22–27, 58; for a review, see 89]. By integrating insights from the dominance/prestige framework of social rank [32–34] and incorporating the inherently contextualized (and often nested) nature of social norms [5, 31, 35, 36], our studies shed new light on when and why norm violators are deemed worthy of influence. The results of Studies 1 and 2 indicate that, in our samples, norm violations increased perceptions of dominance and decreased perceptions of prestige, a pattern that is typically not conducive to voluntary influence granting. However, the results of Studies 3 to 5 indicate that certain constellations of norm violations do enhance prestige and influence granting. Specifically, individuals appear to more readily amass prestige in the eyes of fellow group members, and thereby gain support as leaders, when they violate community norms in order to abide by group norms.

Our findings resonate with previous research on dynamic processes of social hierarchy in the context of nested group structures. A series of studies revealed that contributions in a nested social dilemma had differential consequences for dominance and prestige depending on the structure of the broader intergroup context [42]. Most relevant to the current research is the finding that contributions that benefited the ingroup yet harmed the outgroup increased prestige when harming the outgroup was the only way to benefit the ingroup, but not when there was a possibility to benefit the ingroup without harming the outgroup. Although unrelated to norm violations, the pattern observed by Halevy and colleagues is compatible with the current finding that community norm violations engender more prestige when group and community norms do (rather than do not) conflict and community norm violation is therefore necessary to abide by local group norms.

The current research also forges new links between the dominance/prestige framework and the social dynamics of norm violations. Previous work found that individuals who derived their standing in a group from dominance were penalized more harshly for transgressions than individuals who derived their standing from prestige, because the former were perceived as lacking moral credentials [16]. Although different in its outlook, the conclusion of that study resonates with our finding that norm violators are seen as more prestigious and worthy of leadership in the eyes of fellow group members when they violate community norms in order to abide by group norms than when they violate community norms without an apparent necessity to do so. It is conceivable that transgressors in the former situation would receive

greater moral clearance from their fellow group members for violating community norms as the group stands to benefit from such violations. By identifying conditions under which norm violations increase rather than decrease prestige and influence granting, our findings contribute to a richer understanding of the social mechanics of prestige. Although prestige is typically connected with positive and desirable traits and behaviors, our data point to a potential dark side of prestige, namely that it can be enhanced through behaviors that uphold local group norms yet may undermine global community functioning [see also 90].

Our findings also speak to the fast-growing literature on morality, particularly as it relates to social dynamics within groups [91, 92]. Research has demonstrated that perceived morality is a primary criterion for acceptance in groups [93]. What is deemed moral or immoral is, however, subject to social-contextual influences. For instance, it is well documented that there are differences in the relative valuation of distinct moral foundations across cultures and political ideologies [94]. Although not about morality per se, the present finding that people respond differently to violations of local group norms versus global community norms suggests there is value in further dissecting the social processes underlying morality perceptions and responses to moral violations. It is conceivable that group members who violate community norms to abide by local group norms are perceived as more moral by their fellow group members (though not necessarily by members of the general community; see below), which may further contribute to their appeal as leaders. Furthermore, the acceptability of morally dubious yet ingroup-benefiting behavior may shift as a function of competition with rivaling outgroups [95]. Fellow group members' evaluations of a focal member's norm violations as moral rather than immoral may in turn reinforce future community norm violations that serve to uphold local group norms.

A practical implication of the present research is that when group and community norms are (or appear to be) at odds, individuals can gain standing in the eyes of fellow group members by violating community norms that are not endorsed within the group. Even though people generally perceive norm abiders as more prestigious than norm violators, our data reveal that prestige preferentially tracks abidance by local group norms as opposed to global community norms. As a consequence, in situations where group and community norms conflict, people can gain prestige and influence in groups by prioritizing group norms over community norms. This may help explain how societally disruptive behaviors within certain groups or subcultures (e.g., violence in street gangs, rioting among anti-government protesters) emerge and are perpetuated as perpetrators acquire standing within their groups by violating community norms.

To the degree that people are aware that violating community norms to abide by group norms can bring them social-hierarchical benefits, they may be more likely to strategically and conspicuously violate community norms that are at odds with local group norms when status mobility within their group is perceived as high rather than low. Reflecting the fundamental nature of the desire for status [96], many people are quite attentive to (changes in) status hierarchies and their positions in them [97]. As a result, when a group's hierarchy is subject to change, lower-ranking members may choose to violate community norms that conflict with group norms so as to demonstrate their group credentials. Likewise, non-prototypical or marginalized group members may be motivated to look for ways to prove their group loyalty [98–100], for instance by exhibiting behaviors suggesting they hold local group norms in higher regard than global community norms, particularly if they have a strong desire to be accepted and included in their group [101; also see 102]. When local group norms prescribe behaviors that go against the interests of society at large, such dynamics can result in disruptive patterns of behavior that may be sustained and perpetuated by the status bonus actors receive. Furthermore, for higher-ranking group members, the threat of possible status loss in a hierarchical

system that is in flux may fuel community norm violations as a form of "maintenance work"–to periodically reinforce their standing in the group. In groups that are characterized by a stable hierarchy, in contrast, there would be less of an incentive for high- and low-ranking members alike to strategically violate community norms, because they would not stand to gain much from such violations. If this analysis holds true, community norm violations may be more prevalent in times of change, when hierarchies are perceived as unstable and/or illegitimate, and when people perceive opportunities for advancing their positions in groups.

Our findings also shed new light on various political developments over the past decade. If individuals who abide by the norms of their local group but violate the norms of the community at large indeed gain prestige and influence, as our findings suggest, then violating global community norms that are at odds with local group norms may be a potent strategy to appeal to one's electorate. Indeed, Nigel Farage in the UK, Sebastian Kurz in Austria, Alexander Gauland in Germany, Geert Wilders in the Netherlands, Marine Le Pen in France, and Donald Trump in the US vowed to govern in the interest of their electorate (thus abiding by local group norms), but to break with existing rules of politics (thereby violating overarching norms of the political community; [103, 104]. The success of these politicians may stem, in part, from the prestige they derived among their constituents from the explicit prioritization of local group norms over global community norms.

## Strengths, limitations, and future directions

We employed a variety of methodological approaches in the current research, each of which has its own strengths and limitations. An advantage of the implicit association test employed in Study 1 is that it enabled us to examine generic associations between norm violation versus abidance and dominance versus prestige, which may generalize across different spheres of life. A downside of this approach is that it does not allow for causal conclusions. This limitation was addressed in Study 2, which yielded causal evidence that norm violations can increase dominance and decrease prestige. The scenario approach employed in Study 3 enabled us to manipulate global community norm violations and local group violations orthogonally, thus providing insight into their separate and combined effects. A limitation of this study is that we relied on participants' ability to imagine themselves in the situations we described. This shortcoming was addressed in Study 4, in which we used a recall paradigm to tap into respondents' own memories of norm-violating versus norm-abiding high-school students. Limitations of that study are that we relied on retrospection and that causality cannot be established–issues that are in turn remedied by Studies 3 and 5. Another potential limitation of Study 3 is that we employed an indirect (intersubjective) measure of leadership endorsement to operationalize influence granting. Although this approach alleviates concerns about socially desirable responding [80], it does not provide direct insight into how participants themselves would respond to norm violators versus abiders. To address this limitation, we complemented the intersubjective measure of leadership endorsement with a behavioral measure of leadership task assignment in Study 5. The fact that this behavioral measure produced a very similar pattern of results as the intersubjective measure increases confidence that the intersubjective measure is a valid, albeit more indirect, proxy of actual leadership endorsement. All in all, the limitations of the individual studies notwithstanding, we believe that a strength of the current research is that we found compatible patterns of results across studies involving different methods, operationalizations, and populations.

We used samples from different populations across our studies, each of which comes with their own particularities and constraints. In Studies 1, 2, and 4 we used samples of undergraduate students. Although we deemed these samples suitable in light of the focus and nature of

these studies (general associations between norm violation vs. abidance and dominance vs. prestige, a mundane parking violation, and recollections of high school students' norm violations, respectively), we acknowledge that the rather homogeneous demographic makeup of student samples can limit the generalizability of the results of these studies. We therefore employed samples from different populations in Studies 3 and 5, which we drew from Crowdflower and MTurk, respectively. These platforms offer samples that are much more demographically heterogeneous than student samples [105]. The fact that the findings across these studies and samples are compatible increases our confidence in the generalizability of the conclusions to other populations.

The purpose of the present studies was to establish (causal) effects of different constellations of norm violation versus abidance on dominance, prestige, and influence granting. To this end, we employed experimental approaches, which prioritize internal validity over external validity. Future studies could trade internal validity for external validity to examine whether our findings generalize to more naturalistic field settings besides the school setting of Study 4. A recent analysis of investment banks suggests that the dynamics uncovered here can indeed be meaningfully studied in other real-world settings. Roulet [106] identified a range of behaviors that constitute abidance by local group norms of the banking profession but that are disapproved of on a broader societal level, such as making risky investments, focusing on short-term gains, and giving out large bonuses. This study revealed that corporate customers responded favorably to reports of misconduct by banks in the media, because they interpreted such misconduct as reflecting high service quality. Indeed, the more banks were societally disapproved of for their wrongdoings, the more likely they were to be invited into initial public offering syndicates. In other words, in a context in which local norms of the banking sector clash with global community norms, investment banks that violate community norms are preferred by investors over banks that abide by community norms. More research in such naturalistic settings may further establish the real-world implications of different constellations of norm violations for social-hierarchical dynamics.

Our interest in the current research was in illuminating the effects of norm violations on perceptions of dominance and prestige, and the downstream consequences of such perceptions for influence granting. Accordingly, we built part of our theorizing on the dominance/prestige framework of social rank [32–34], and focused on dominance and prestige as possible mediators of the effects of (counter)normative behavior on influence granting. Although a considerable body of empirical work supports the notion that dominance and prestige are distinct routes to influence [e.g., 32, 42, 43, 49, 107–110], it is worth noting that the distinction can become blurry in particular circumstances. For instance, traits or characteristics that are associated with dominance (e.g., physical strength) are sometimes also associated with expertise relevant for the survival of the group (e.g., conflict resolution; e.g., [108, 110]). In such cases, it may be more difficult to identify separate pathways via dominance and prestige. Moreover, even though we found consistent support for a mediating role of prestige, it is possible that other mechanisms are at play that we did not consider. One such mechanism may be perceived loyalty. It is conceivable that individuals who prioritize local group norms over global community norms when the two conflict are perceived as loyal by fellow group members, which may increase their leadership appeal. Given that prestige tracks individuals' perceived value for groups, we see this interpretation as compatible with the observed mediation through prestige–that is, a person's perceived loyalty may shape their prestige in the eyes of fellow group members, which in turn would lead those group members to grant influence to that person. Future research could examine whether the tendency to confer prestige to individuals who prioritize local group norms over global community norms is mediated by perceived loyalty.

It is worth noting that the focus in three of our five studies was on rather mundane norm violations: faulty parking in Study 2, violation of a dress code in Study 3, and violation of a company formatting rule in Study 5. Although the compatible results obtained across these different contexts and behaviors enhance confidence in the generalizability of the findings, it is possible that our conclusions are limited to relatively inconsequential norm violations. The correlational patterns observed in Studies 1 and 4 may be based on a broader range of violations, including potentially more consequential ones, but our data do not allow us to examine this. Future research could investigate whether similar patterns hold for more serious norm violations, although we note that ethical constraints may make such studies challenging to conduct.

In the current research we focused on different constellations of abidance and violation of norms of local groups that were nested in global communities. Although such nesting is characteristic of many group structures in daily life (e.g., classes within schools, teams within organizations, parties within parliament), other contextual factors are also worth investigating. One aspect that seems particularly promising is the degree to which different types of norms are valued. It stands to reason that people value the norms of their local groups more than those of the global communities in which those groups are embedded, because to the degree that they identify with their local groups, endorsing local group norms is a way to give expression to their social identity [40]. If true, the relative valuation of different sets of social norms may moderate responses to (counter)normative behavior also in the context of other types of normative tension. In particular, when faced with a conflict between different social norms, people can be expected to amass greater prestige and influence when they prioritize abidance by norms that are deemed more rather than less important by fellow group members.

Another open question is what role, if any, the observer's group membership plays in shaping responses to norm violators. We know from previous work that deviant group members are often looked down upon, in part because they undermine the positive social identity fellow group members can derive from their group membership [17, 57, 65, 66]. Against this background, it is possible that the favorable responses to individuals who prioritize local group norms over global community norms observed in the current research are stronger for ingroup observers than for uninvolved observers or members of the outgroup. Such a pattern would be consistent with the theoretical argument that social norms are contextualized [5, 31] and that prestige has evolved as a mechanism to highlight individuals who possess attributes that generate benefits for the group [33]. It is possible, furthermore, that the enhanced prestige of community norm violators in the eyes of their ingroup goes hand in hand with reduced prestige in the eyes of other members of society. Our data are not conclusive in this regard. On the one hand, Studies 1 and 2 indicate that norm violations generally reduce prestige, which speaks to the risk of reduced prestige of community norm violators outside their immediate group. On the other hand, participants in Study 5 conferred prestige to and selected leadership tasks for a community norm violator they did not know and therefore most likely did not consider part of their ingroup. Future research could investigate more directly in whose eyes community norm violators gain or lose prestige.

We focused the present investigation on groups nested within larger communities, because norm violations in such social constellations are particularly impactful as they have implications for members of the community at large and can undermine the normative systems that hold societies together. Moreover, conflicts between different sets of norms are likely to be particularly salient when groups are nested, as it is not possible to "escape" the norms of the global community because the local group is a part of it. Even so, we believe it would be worthwhile to expand the scope of the current investigation beyond nested social groups by examining the social consequences of norm violations in situations where individuals are faced with conflicting norms between non-nested groups (e.g., family vs. friends). Based on our current

theorizing and findings, we would predict that individuals who violate the norms of one group in order to abide by the competing norms of another group preferentially gain status in that latter group, while their status in the former group may be reduced. Future research could investigate the social consequences of different ways in which people may navigate tensions between norms in different groups they are a part of.

As noted in the Introduction, prior research in the realm of the dominance/prestige framework has typically found that prestige is conducive to voluntary influence granting, whereas dominance is not [50]. Our findings are in line with that pattern. However, previous work has also identified specific circumstances under which dominance does become a positive predictor of leadership endorsement, including economic uncertainty [43] and intergroup competition [46–49]. In light of such findings, it stands to reason that community norm violators who abide by the norms of their local group are particularly likely to gain influence under harsh conditions such as intergroup conflict or economic recession, because their dominance and prestige in the eyes of others would work in tandem to increase their leadership appeal. Historical observations of the political trajectories of certain infamous leaders provide suggestive support for this possibility, which could be investigated in future research. In addition, future work could further examine the effects of norm violation on dominance. It is conceivable that these effects are also subject to moderating factors, such as characteristics of the perpetrator (e.g., formal power, physical formidability) or the magnitude of possible sanctions for the violation [25].

Finally, the current studies were conducted in independent cultures (the US and the Netherlands) in which norm violations are deemed comparatively more acceptable than in interdependent cultures [10]. Future research could expand the scope of the current investigation by comparing the effects of community and group norm violations in interdependent versus independent cultures. One possible prediction is that individuals who violate global community norms to abide by local group norms are appreciated even more in interdependent cultures because they signal that they give priority to the interests of the group. Alternatively, community norm violators may be frowned upon because their behavior jeopardizes societal functioning and harmony, notions that are valued comparatively more in interdependent cultures [111]. Moreover, it is conceivable that the patterns we uncovered here are modulated by other aspects of the socio-economic system within which norm violations occur, such as egalitarianism versus social stratification and the role the violated norm plays in society [112]. Cross-cultural investigations of the effects of global community norm violations versus local group norm violations could further enhance understanding of when norm violators gain influence.

## Conclusion

The present studies provide insight in whether, when, and why norm violators gain influence in groups. We discovered that norm violators are perceived as rather high on dominance and low on prestige, a pattern that typically undermines voluntary influence granting. However, individuals who violate global community norms so as to abide by local group norms are perceived as comparatively more prestigious, which increases their leadership appeal. Thus, whereas people are generally reluctant to voluntarily grant influence to individuals who violate community norms, they are willing to grant influence to those who violate community norms in order to uphold group norms–that is, to rebels with a cause.

## Supporting information

**S1 File. Supplemental material.** Additional methodological details, data, and analyses. (DOCX)

**S1 Table. Original Dutch target words employed in the implicit association test (IAT) together with English translations (Study 1).**
(DOCX)

**S2 Table. The extent to which a popular and a typical (control) student violated norms of teachers, other students, and peers (Study 3).**
(DOCX)

**S3 Table. Indirect effects of condition on the assignment of leadership tasks via prestige and dominance as estimated by three different imputation methods (Study 5).**
(DOCX)

**S4 Table. Indirect effects of condition on the assignment of leadership tasks via prestige as estimated by three different imputation methods (Study 5).**
(DOCX)

**S1 Fig. Multiple mediation model testing the effect of conflict between local group norms and global community norms on assignment of leadership tasks via prestige and dominance after imputing missing values on the dependent variable via stochastic imputation (Study 5).**
(TIF)

**S2 Fig. Multiple mediation model testing the effect of conflict between local group norms and global community norms on assignment of leadership tasks via prestige and dominance after imputing missing values on the dependent variable via predictive mean matching (Study 5).**
(TIF)

## Author Contributions

**Conceptualization:** Gerben A. van Kleef, Florian Wanders, Annelies E. M. van Vianen, Astrid C. Homan.

**Formal analysis:** Florian Wanders, Rohan L. Dunham, Xinkai Du.

**Investigation:** Florian Wanders, Xinkai Du.

**Methodology:** Gerben A. van Kleef, Florian Wanders, Xinkai Du, Astrid C. Homan.

**Project administration:** Gerben A. van Kleef, Florian Wanders.

**Software:** Florian Wanders.

**Supervision:** Gerben A. van Kleef, Annelies E. M. van Vianen, Astrid C. Homan.

**Visualization:** Rohan L. Dunham.

**Writing – original draft:** Gerben A. van Kleef, Florian Wanders, Annelies E. M. van Vianen, Astrid C. Homan.

**Writing – review & editing:** Gerben A. van Kleef, Florian Wanders, Annelies E. M. van Vianen, Astrid C. Homan.

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
