## [Decision Letter · Decision Letter 0]

21 Aug 2023

PONE-D-23-23034Rebels with a Cause? How Norm Violations Shape Dominance, Prestige, and Influence GrantingPLOS ONE

Dear Dr. van Kleef,

Thank you for submitting your manuscript to PLOS ONE. After careful consideration, we feel that it has merit but does not fully meet PLOS ONE’s publication criteria as it currently stands. Therefore, we invite you to submit a revised version of the manuscript that addresses the points raised during the review process.

We look forward to receiving your revised manuscript.

Kind regards,

Anastassia Zabrodskaja, Ph.D.

Academic Editor

PLOS ONE

Journal Requirements:

Additional Editor Comments:

Please read all reviewers' comments and revise your paper carefully. 

Reviewers' comments:

Reviewer's Responses to Questions

**Comments to the Author**

1. Is the manuscript technically sound, and do the data support the conclusions?

Reviewer #1: Yes

Reviewer #2: Yes

2. Has the statistical analysis been performed appropriately and rigorously? 

Reviewer #1: Yes

Reviewer #2: Yes

3. Have the authors made all data underlying the findings in their manuscript fully available?

Reviewer #1: No

Reviewer #2: No

4. Is the manuscript presented in an intelligible fashion and written in standard English?

Reviewer #1: Yes

Reviewer #2: Yes

5. Review Comments to the Author

Reviewer #1: Thank you for the opportunity to review "Rebels with a Cause? How Norm Violations Shape Dominance, Prestige, and Influence Granting". In this manuscript, the authors examine the effect of norm violations on status conferral. The manuscript is well written, the experiments are well designed, and the manuscript concludes with insightful predictions for future research. My concerns are minimal, but I would appreciate the authors providing clarity in their response letter on the following points.

The author’s argument is that individuals gain prestige when they violate global norms in order to comply with local norms. And the theoretical underpinning is that by complying with the local norm at the expense of the global norm, the violator is engaging is a costly act of support for the group, and as a result garners prestige. The authors provide support for this by showing that members of the local group (or participants answering from the perspective of someone from the local group) rate the violator as more prestigious in such contexts.

However, might not members of the global community have a very different perspective of such violators? I doubt they would also grant prestige to such violators. So, is the argument actually more nuanced: individuals gain prestige from their local group and lose prestige from their global group when they violate global norms in order to comply with local norms? And on net wouldn’t the violator actually lose prestige given that the global group consists of more people than the local group? I suppose prestige from one’s local group is particularly beneficial, so much so that it sufficiently offsets the decline in prestige amongst the broader community, making such behaviour adaptive.

Relatedly, is this really about a comparison of global versus local, or is it just a comparison of allegiance to one group as opposed to another group? For example, wouldn’t the same effect be observed if one chose to attend their football club’s match, thereby forgoing attendance at their rugby club’s match. The violator would gain prestige with their football club while losing prestige with their rugby club. Is the local versus global comparison necessary for this effect to occur? If the global versus local comparison is necessary, does the scope of comparison matter? For example, imagine that nested within a local group was a micro group. Would breaching the micro group’s norms for the sake of the local group’s norms result in a gain or loss of prestige? And what would each outcome imply for the authors’ theory?

The authors demonstrate in Study 5 that what is key for garnering prestige is that the compliance with the local norm requires the violation of the global norm. In other words, the violator needs to be a ‘rebel with a cause’ rather than a rebel without one. If that is the case, I am curious to know whether the authors see the results from Study 4 fitting with this argument. Were the popular students in Study 4 violating the teacher’s norms in order to comply with the norms of their peers? Or were they simply violating the teacher’s norms in order to be rebellious (without a cause)? I realise that this was a recall study which doesn’t allow for the same level of precision on such matters, but I would appreciate the authors to speculate on this point as it does seem central to their theorizing.

Typo on bottom of Page 24: “…rated the protagonist as les[s] prestigious…”

Reviewer #2: I commend the authors on producing a well-written manuscript built on sound empirical studies. I believe this has the potential to be an important contribution across a range of disciplines.

Overview:

This study focuses on understanding the relationship between norm violations and influence within social groups. The authors explore how norm violators either gain or lose influence via variation in their perceived dominance and prestige. They found that norm violations are linked to perceptions of dominance, while norm abidance is associated with prestige. The authors conducted a series of integrated studies, including experimental designs, demonstrating that when local group norms and global community norms conflict, individuals who violate global norms but adhere to local norms are perceived as both dominant and prestigious, leading to greater influence and endorsement as leaders. This research sheds light on the complex interplay of norms, influence, and leadership emergence within social groups and variation within and between social groups.

The authors are correct in their assertion that much of the literature on norm violations and influence overlooks the importance of social context and nested and overlapping social groups. The authors make an important contribution in emphasizing this point and incorporating it into their empirical frameworks.

I find the methods appropriate and results convincing.

My main critics are 1) I would prefer if the authors would have better conformed to open science practices, including preregistering predictions and analyses. Perhaps I missed it, but I also did not see that data and code are openly available. I think the data and code are only available upon publication. Therefore, I cannot evaluate “behind the scenes”. I realize, however, that the data were collected several years ago. The authors could be more straight forward about when or why these practices were not followed and if any elements of “researcher degrees of freedom” influenced processes or decisions. And, 2) I would prefer to see more discussion about the limitations of the sample, the particularities of the samples, and generalizability, and how the sample characteristics constrain what these results can tell us about “human” cognition or social behavior.

There is another relevant idea I did not see authors discuss. I’m thinking about, when high status, prestigious leaders are able to violate certain norms because of their high status and prestige. A common example is ethnography includes high status big men in horticultural societies being able to strategically violate marriage norms in finding/taking a wife. In such cases, the community is aware that they have violated a norm and the community has available mechanisms to punish the leader for the norm violation. But, because of their history of prosocial investments and good leadership, these sorts of global norm violations may be overlooked. There could be other more local norm violations as well. I suppose, a core point of this idea is that naturalistic individual behavior often does not exist in a one-shot interaction (and this was probably rarely the case over human evolutionary history). Therefore, a reputation and history of prestigious prosociality may afford an individual the right to violate particular norms in certain contexts.

See:

Lewis, H. S. (1974). Leaders and followers: Some anthropological perspectives. Addison-Wesley.

Boehm, C. (1999). Hierarchy in the forest: The evolution of egalitarian behavior. Harvard University Press Cambridge, MA.

Chagnon, N. A. (1968). Yaomamö, the fierce people. Holt, Rinehart and Winston.

Other comments:

Line 149: It may be worth mentioning that the degree to which, and/or when dominance and prestige are more or less distinct has been challenged on both theoretical and empirical grounds.

See:

Von Rueden, C. (2014). The roots and fruits of social status in small-scale human societies. In J. T. Cheng, J. L. Tracy, & C. Anderson (Eds.), The psychology of social status (pp. 179–200). Springer.

von Rueden, C., Gurven, M., Kaplan, H., & Stieglitz, J. (2014). Leadership in an Egalitarian Society. Hum Nat, 25(4), 538–566. https://doi.org/10.1007/s12110-014-9213-4

Chapais, B. (2015). Competence and the Evolutionary Origins of Status and Power in Humans. Hum Nat, 26, 161–183. https://doi.org/10.1007/s12110-015-9227-6

Jiménez, Á. V., & Mesoudi, A. (2019). Prestige-biased social learning: Current evidence and outstanding questions. Palgrave Communications, 5(1), 20. https://doi.org/10.1057/s41599-019-0228-7

Garfield, Z. H., & Hagen, E. H. (2020). Investigating evolutionary models of leadership among recently settled Ethiopian hunter-gatherers. Special Issue on Evolution and Biology of Leadership, 31(2), 101290. https://doi.org/10.1016/j.leaqua.2019.03.005

Line 159-162: An important nuance, however, includes that followers may prefer dominant leaders and preferentially select them as leaders in the context of cooperative activities, given their superior abilities to deter free riders. Therefore, dominance can be an effective leadership strategy/suit of characteristics for promoting followership.

See:

von Rueden, C. R. (2022). Unmaking egalitarianism: Comparing sources of political change in an Amazonian society. Evolution and Human Behavior. https://doi.org/10.1016/j.evolhumbehav.2022.09.001

Line 165-166: This trend also appears to be widespread across human societies.

See:

Lewis, H. S. (1974). Leaders and followers: Some anthropological perspectives. Addison-Wesley.

Garfield, Z. H., Hubbard, H., Robert, & Hagen, E. H. (2019). Evolutionary models of leadership: Tests and synthesis. Human Nature, 30(1), 23–58. https://doi.org/10.1007/s12110-019-09338-4

Line 168: The authors are correct that the social learning theory of the evolution of prestige has been largely influential, however, it may be worth noting there are alternative theoretical frameworks developed in recent years, particularly those focused on benefit generation and cost infliction capacities.

See:

Von Rueden, C. (2014). The roots and fruits of social status in small-scale human societies. In J. T. Cheng, J. L. Tracy, & C. Anderson (Eds.), The psychology of social status (pp. 179–200). Springer.

Pietraszewski, D. (2019). The evolution of leadership: Leadership and followership as a solution to the problem of creating and executing successful coordination and cooperation enterprises. The Leadership Quarterly. https://doi.org/10.1016/j.leaqua.2019.05.006

Hagen, E. H., & Garfield, Z. (2019). Leadership and prestige, mothering, sexual selection, and encephalization: The computational services model [Preprint]. Open Science Framework. https://doi.org/10.31219/osf.io/9bcdk

Garfield, Z. H., Syme, K. L., & Hagen, E. H. (2020). Universal and variable leadership dimensions across human societies. Evolution and Human Behavior, 41(5), 397–414. https://doi.org/10.1016/j.evolhumbehav.2020.07.012

Durkee, P. K., Lukaszewski, A. W., & Buss, D. M. (2020). Psychological foundations of human status allocation. Proceedings of the National Academy of Sciences, 117(35), 21235–21241. https://doi.org/10.1073/pnas.2006148117

Line 199-201: Is this necessarily true? People may violate norms unintentionally. Or, they may have some prior expectation on the likelihood their violations will be met with sanctions. And based on other contextual factors (which the authors review well) they may “compute” some set of cost-benefit analyses on the probability of backlash and the severity of any potential backlash they may be subjected to.

Line 200-203: I would also suggest the embodied capital/wealth inequality model is important here, i.e., individuals well endowed with certain forms of capital, e.g., social, material, or embodied capital, may be able to withstand or recuperate from certain forms of backlash.

See:

Von Rueden, C. (2014). The roots and fruits of social status in small-scale human societies. In J. T. Cheng, J. L. Tracy, & C. Anderson (Eds.), The psychology of social status (pp. 179–200). Springer.

Mattison, S. M., Smith, E. A., Shenk, M. K., & Cochrane, E. E. (2016). The evolution of inequality. Evolutionary Anthropology: Issues, News, and Reviews, 25, 184–199.

Lancaster, J. B., & Kaplan, H. S. (2010). Embodied Capital and Extra-somatic Wealth in Human Evolution and Human History. In M. P. Muehlenbein (Ed.), Human Evolutionary Biology (pp. 439–456). New York: Cambridge University Press.

Introduction, generally: The authors points about local and group norm norms and nested social groups is compelling and important. For recent and consistent cross-cultural work highlighting the importance of socio-ecology in shaping norm violations see:

Garfield, Z. H., Ringen, E. J., Buckner, W., Medupe, D., Wrangham, R. W., & Glowacki, L. (2023). Norm violations and punishments across human societies. Evolutionary Human Sciences, 5, e11. https://doi.org/10.1017/ehs.2023.7

Line 298: Were the predictions and/or methods preregistered? If not, can the authors justify or discuss this decision?

General comment: The framework the authors have developed (and used in their predictions) seems to rely on the assumption of distinctiveness between dominance and prestige, which is fair. But the authors also explain well that nested groups and norms overlap and can contradict. The same principle applies to the conceptions and representations of dominance and prestige, as the authors have defined them. For example, if "aggression, coercion, and intimidation" are seen as locally valued "skills, expertise, or abilities", would the expression and embodiment of these qualities be defined as dominance or prestige? This seems to be important concerning the hypotheses to be tested.

Study 1: Perhaps I missed it, but were participants given any explanation or definitions of dominance and prestige?

General discussion:

The authors have done well to discuss their samples and participants. However, I feel they could/should discuss the limitations and generalizability in greater detail. Particularly in drawing on theoretical frameworks rooted in evolutionary theory (i.e., the dominance-prestige framework as developed by the cited authors). There are a few minor instances of fairly broad language in the author’s discussions. For example, “across the board” (Line 1192) and “individuals” (Line 1196). I realize that a diligent reader will not make the mistake of over generalizing the author’s interpretations. But, as a safeguard and in respect with current best practices, I suggest discussing the potential limits to generalizability, based on what is known about the participant sample. The cultural tightness-looseness or WEIRD people frameworks might be useful. That is, given the samples of students, etc., what can the authors say about their results and generalizability, in a bit more detail.

For detailed discussions of external validity and ways to discuss in a comprehensive fashion see:

Findley, M. G., Kikuta, K., & Denly, M. (2021). External Validity. Annual Review of Political Science, 24(1), 365–393. https://doi.org/10.1146/annurev-polisci-041719-102556

6. PLOS authors have the option to publish the peer review history of their article (what does this mean?). If published, this will include your full peer review and any attached files.

Reviewer #1: No

Reviewer #2: No

---

## [Author Response · Author response to Decision Letter 0]

4 Oct 2023

To preserve formatting, we have uploaded a separate file with detailed responses to the Editor's and the Reviewers' comments.

---

## [Decision Letter · Decision Letter 1]

25 Oct 2023

Rebels with a Cause? How Norm Violations Shape Dominance, Prestige, and Influence Granting

PONE-D-23-23034R1

Dear Dr. van Kleef,

We’re pleased to inform you that your manuscript has been judged scientifically suitable for publication and will be formally accepted for publication once it meets all outstanding technical requirements.

Kind regards,

Anastassia Zabrodskaja, Ph.D.

Academic Editor

PLOS ONE

Additional Editor Comments (optional):

Reviewers' comments:

Reviewer's Responses to Questions

**Comments to the Author**

1. If the authors have adequately addressed your comments raised in a previous round of review and you feel that this manuscript is now acceptable for publication, you may indicate that here to bypass the “Comments to the Author” section, enter your conflict of interest statement in the “Confidential to Editor” section, and submit your "Accept" recommendation.

Reviewer #1: All comments have been addressed

2. Is the manuscript technically sound, and do the data support the conclusions?

Reviewer #1: Yes

3. Has the statistical analysis been performed appropriately and rigorously? 

Reviewer #1: Yes

4. Have the authors made all data underlying the findings in their manuscript fully available?

Reviewer #1: Yes

5. Is the manuscript presented in an intelligible fashion and written in standard English?

Reviewer #1: Yes

6. Review Comments to the Author

Reviewer #1: (No Response)

7. PLOS authors have the option to publish the peer review history of their article (what does this mean?). If published, this will include your full peer review and any attached files.

Reviewer #1: No

---

## [Editor Report · Acceptance letter]

13 Nov 2023

PONE-D-23-23034R1 

Rebels with a Cause? How Norm Violations Shape Dominance, Prestige, and Influence Granting 

Dear Dr. van Kleef:

I'm pleased to inform you that your manuscript has been deemed suitable for publication in PLOS ONE. Congratulations! Your manuscript is now with our production department. 

Kind regards, 

on behalf of

Professor Anastassia Zabrodskaja 

Academic Editor

PLOS ONE